# Fused Gromov-Wasserstein Graph Mixup for Graph-level Classifications

**Xinyu Ma**[1]    **Xu Chu**[2][*]  **Yasha Wang**[1,3]    **Yang Lin**[1]    **Junfeng Zhao**[1]
**Liantao Ma**[1,3]    **Wenwu Zhu**[2]

[1]School of Computer Science, Peking University
[2]Department of Computer Science and Technology, Tsinghua University
[3]National Research and Engineering Center of Software Engineering, Peking University
`maxinyu@pku.edu.cn, chu_xu@mail.tsinghua.edu.cn`

## Abstract

Graph data augmentation has shown superiority in enhancing generalizability and robustness of GNNs in graph-level classifications. However, existing methods primarily focus on the augmentation in the graph signal space and the graph structure space independently, neglecting the joint interaction between them. In this paper, we address this limitation by formulating the problem as an optimal transport problem that aims to find an optimal inter-graph node matching strategy considering the interactions between graph structures and signals. To solve this problem, we propose a novel graph mixup algorithm called `FGWMixup`, which seeks a "midpoint" of source graphs in the Fused Gromov-Wasserstein (FGW) metric space. To enhance the scalability of our method, we introduce a relaxed FGW solver that accelerates `FGWMixup` by improving the convergence rate from $\mathcal{O}(t^{-1})$ to $\mathcal{O}(t^{-2})$. Extensive experiments conducted on five datasets using both classic (MPNNs) and advanced (Graphormers) GNN backbones demonstrate that `FGWMixup` effectively improves the generalizability and robustness of GNNs. Codes are available at `https://github.com/ArthurLeoM/FGWMixup`.

## 1   Introduction

In recent years, Graph Neural Networks (GNNs) [1, 2] have demonstrated promising capabilities in graph-level classifications, including molecular property prediction [3, 4], social network classification [5], healthcare prediction [6, 7], etc. Nevertheless, similar to other successfully deployed deep neural networks, GNNs also suffer from data insufficiency and perturbation, requiring the application of regularization techniques to improve generalizability and robustness [8]. Data augmentation is widely adopted for regularization in deep learning. It involves creating new training data by applying various semantic-invariant transformations to the original data, such as cropping or rotating images in computer vision [9], randomly inserting and rephrasing words in natural language [10, 11], etc. The augmented data fortify deep neural networks (DNNs) against potential noise and outliers underlying insufficient samples, enabling DNNs to learn more robust and representative features.

Data augmentation for GNNs requires a unique design due to the distinctive properties of attributed graphs [12], such as irregular sizes, misaligned nodes, and diverse topologies, which are not encountered when dealing with data in the Euclidean spaces such as images and tabular data. Generally, the augmentation for GNNs necessitates the consideration of two intertwined yet complementary input spaces, namely the graph signal space $\mathcal{X}$ and the graph structure space $\mathcal{A}$, which are mapped to an aligned latent space $\mathcal{H}$ with GNNs. The graph signal space $\mathcal{X}$ consists of node features. Current

---

[*]Corresponding Author

37th Conference on Neural Information Processing Systems (NeurIPS 2023).

GNNs rely on parameterized nonlinear transformations to process graph signals, which can be efficiently encoded and serve as crucial inputs for downstream predictions. Therefore, the augmentation of graph signals is significant for regularizing the GNN parameter space. On the other hand, the graph structure space $\mathcal{A}$ consists of information about graph topology. Traditional Message Passing Neural Networks (MPNNs), such as GCN and GIN, perform feature aggregation based on edge connectivity. Great efforts [13, 14, 15, 16] have been further exerted on enhancing the expressive power of GNNs [17], which carry out new GNN architectures with stronger sensitivity to graph topology compared with MPNNs. Hence, a good augmentation method should also consider the graph structure space.

Recently, huge efforts have been made to design graph data augmentation methods based on the two spaces. Mainstream research [18, 19, 20, 21, 22] considers the augmentation in the graph signal space and the graph structure space **independently**. For instance, ifMixup [19] conducts Euclidean mixup in the graph signal space, yet fails to preserve key topologies of the original graphs. $\mathcal{G}$-Mixup [20] realizes graph structure mixup based on the estimated graphons, yet fails to assign semantically meaningful graph signals. In fact, the graph signal and structure spaces are not isolated from each other, and there are strong entangling relations between them [23]. Therefore, **a joint modeling of the interaction between the two spaces is essential for conducting graph data augmentation** (joint modeling problem for short).

Aiming at graph data augmentation and addressing the joint modeling problem, we design a novel graph mixup [24] method considering the interaction of the two spaces during the mixup procedure. The key idea is to solve a proper inter-graph node matching strategy in a metric space that measures the distance with respect to both graph signals and structures. We propose to compute the distance metric by solving an optimal transport (OT) problem [25]. The OT problem solves the optimal coupling between nodes across graphs in the fused Gromov-Wasserstein metric space [26], wherein the distance between points takes the interaction between the graph signals and structures into account. Specifically, following [26], graphs can be modeled as probability distributions embedded in the product metric space $\mathcal{X} \times \mathcal{A}$. Our objective is to solve an augmented graph that minimizes the weighted sum of transportation distances between the distributions of the source graphs and the objective graph in this metric space. Developing from the Gromov-Wasserstein metric [27], Fused Gromov-Wasserstein (FGW) distance [26] has been designed to calculate the transportation distance between two unregistered probability distributions defined on different product metric spaces comprising two components, such as graph signals and structures, which defines a proper distance metric for attributed graphs. In short words, the solved graph is the augmented mixup graph in a space considering the interaction between the graph signals and structures.

However, trivially adopting FGW distance solvers [28, 26] is not scalable to large graph datasets due to a heavy computation burden. The computational bottleneck of FGW-based solvers is the nested triple-loop optimization [29], mainly due to the polytope constraint for the coupling matrix in the FGW solver. Inspired by [30], we disentangle the polytope constraint into two simplex constraints for rows and columns respectively, thence executing mirror descent with projections on the two simplexes in an alternating manner to approximate the original constraint. We prove that with a bounded gap with the ideal optimum, we may optimize the entire algorithm into a double-loop structure at convergence rate $\mathcal{O}(t^{-2})$, improving the $\mathcal{O}(t^{-1})$ rate of traditional FGW solvers.

In summary, we highlight the contributions of this paper: We address the challenge of enhancing the generalizability and robustness of GNNs by proposing a novel graph data augmentation method that models the interaction between the graph signal and structure spaces. We formulate our objective as an OT problem and propose a novel graph mixup algorithm dubbed `FGWMixup` that seeks a "midpoint" of two graphs defined in the graph structure-signal product metric space. We employ FGW as the distance metric and speed up `FGWMixup` by relaxing the polytope constraint into disentangled simplex constraints, reducing the complexity from nested triple loops to double loops and meanwhile improve the convergence rate of `FGWMixup`. Extensive experiments are conducted on five datasets and four classic (MPNNs) and advanced (Graphormers) backbones. The results demonstrate that our method substantially improves the performance of GNNs in terms of their generalizability and robustness.

## 2 Methodology

In this section, we formally introduce the proposed graph data augmentation method dubbed `FGWMixup`. We first introduce Fused Gromov-Wasserstein (FGW) distance that presents a distance metric between graphs considering the interaction of graph signal and structure spaces.

Then we propose our optimization objective of graph mixup based on the FGW metric and the algorithmic solutions. Finally, we present our acceleration strategy. In the following we denote $\Delta_n := \{\boldsymbol{\mu} \in \mathbb{R}_+^n | \sum_i \mu_i = 1\}$ as the probability simplex with $n$-bins, and $\mathbb{S}_n(\mathbb{A})$ as the set of symmetric matrices of size $n$ taking values in $\mathbb{A} \subset \mathbb{R}$.

## 2.1 Fused Gromov-Wasserstein Distance

In OT problems, an undirected attributed graph $G$ with $n$ nodes is defined as a tuple $(\boldsymbol{\mu}, \boldsymbol{X}, \boldsymbol{A})$. $\boldsymbol{\mu} \in \Delta_n$ denotes the probability measure of nodes within the graph, which can be modeled as the relative importance weights of graph nodes. Common choices of $\boldsymbol{\mu}$ are uniform distributions [31] ($\boldsymbol{\mu} = \mathbf{1}_n/n$) and degree distributions [32] ($\boldsymbol{\mu} = [\deg(v_i)]_i/(\sum_i \deg(v_i))$ ). $\boldsymbol{X} = (\boldsymbol{x}^{(1)}, \cdots, \boldsymbol{x}^{(n)})^\top \in \mathbb{R}^{n \times d}$ denotes the node feature matrix with $d$-dimensional feature on each node. $\boldsymbol{A} \in \mathbb{S}_n(\mathbb{R})$ denotes a matrix that encodes structural relationships between nodes, which can be selected from adjacency matrix, shortest path distance matrix or other distance metrics based on the graph topologies. Given two graphs $G_1 = (\boldsymbol{\mu_1}, \boldsymbol{X_1}, \boldsymbol{A_1})$ and $G_2 = (\boldsymbol{\mu_2}, \boldsymbol{X_2}, \boldsymbol{A_2})$ of sizes $n_1$ and $n_2$ respectively, Fused Gromov-Wasserstein distance can be defined as follows:

$$\text{FGW}_q(G_1, G_2) = \min_{\boldsymbol{\pi} \in \Pi(\boldsymbol{\mu_1}, \boldsymbol{\mu_2})} \sum_{i,j,k,l} \left( (1 - \alpha) d\left( \boldsymbol{x}_1^{(i)}, \boldsymbol{x}_2^{(j)} \right)^q + \alpha \left| \boldsymbol{A_1}(i,k) - \boldsymbol{A_2}(j,l) \right|^q \right) \pi_{i,j} \pi_{k,l},$$
(1)

where $\Pi(\boldsymbol{\mu_1}, \boldsymbol{\mu_2}) := \{\boldsymbol{\pi} \in \mathbb{R}_+^{n_1 \times n_2} | \boldsymbol{\pi} \mathbf{1}_{n_2} = \boldsymbol{\mu_1}, \boldsymbol{\pi}^\top \mathbf{1}_{n_1} = \boldsymbol{\mu_2}\}$ is the set of all valid couplings between node distributions $\boldsymbol{\mu_1}$ and $\boldsymbol{\mu_2}$, $d(\cdot, \cdot)$ is the distance metric in the feature space, and $\alpha \in [0, 1]$ is the weight that trades off between the Gromov-Wasserstein cost on graph structure and Wasserstein cost on graph signals. The FGW distance is formulated as an optimization problem that aims to determine the optimal coupling between nodes in a fused metric space considering the interaction of graph structure and node features. The optimal coupling matrix $\boldsymbol{\pi}^*$ serves as a soft node matching strategy that tends to match two pairs of nodes from different graphs that have both similar graph structural properties (such as $k$-hop connectivity, defined by $\boldsymbol{A}$) and similar node features (such as Euclidean similarity, defined by $d(\cdot, \cdot)$). In fact, FGW also defines a strict distance metric on the graph space $(\Delta, \mathcal{X}, \mathcal{A})$ when $q = 1$ and $\boldsymbol{A}$s are distance matrices, and defines a semi-metric whose triangle inequality is relaxed by $2^{q-1}$ for $q > 1$. In practice, we usually choose $q = 2$ and Euclidean distance for $d(\cdot, \cdot)$ to calculate FGW distance.

**Solving FGW Distance**   Solving FGW distance is a non-convex optimization problem, whose non-convexity comes from the quadratic term of the GW distance. There has been a line of research contributing to solving this problem. [26] proposes to apply the conditional gradient (CG) algorithm to solve this problem, and [28] presents an entropic approximation of the problem and solves the optimization through Mirror Descent (MD) algorithm according to KL-divergence.

## 2.2 Solving Graph Mixup in the FGW Metric Space

Building upon the FGW distance and its properties, we propose a novel graph mixup method dubbed `FGWMixup`. Formally, our objective is to solve a synthetic graph $\tilde{G} = (\tilde{\boldsymbol{\mu}}, \tilde{\boldsymbol{X}}, \tilde{\boldsymbol{A}})$ of size $\tilde{n}$ that minimizes the weighted sum of FGW distances between $\tilde{G}$ and two source graphs $G_1 = (\boldsymbol{\mu_1}, \boldsymbol{X_1}, \boldsymbol{A_1})$ and $G_2 = (\boldsymbol{\mu_2}, \boldsymbol{X_2}, \boldsymbol{A_2})$ respectively. The optimization objective is as follows:

$$\arg \min_{\tilde{G} \in (\Delta_{\tilde{n}}, \mathbb{R}^{\tilde{n} \times d}, \mathbb{S}_{\tilde{n}}(\mathbb{R}))} \lambda \text{FGW}(\tilde{G}, G_1) + (1 - \lambda) \text{FGW}(\tilde{G}, G_2),$$
(2)

where $\lambda \in [0, 1]$ is a scalar mixing ratio, usually sampled from a Beta$(k, k)$ distribution with hyperparameter $k$. This optimization problem formulates the graph mixup problem as an OT problem that aims to find the optimal graph $\tilde{G}$ at the "midpoint" of $G_1$ and $G_2$ in terms of both graph signals and structures. When the optimum $\tilde{G}^*$ is reached, the solutions of FGW distances between $\tilde{G}^*$ and the source graphs $G_1, G_2$ provide the node matching strategies that minimize the costs of aligning graph structures and graph signals. The label of $\tilde{G}$ is then assigned as $y_{\tilde{G}} = \lambda y_{G_1} + (1 - \lambda) y_{G_2}$.

In practice, we usually fix the node probability distribution of $\tilde{G}$ with a uniform distribution (i.e., $\tilde{\boldsymbol{\mu}} = \mathbf{1}_{\tilde{n}}/\tilde{n}$) [26]. Then, Eq.2 can be regarded as a nested bi-level optimization problem, which composes the upper-level optimization *w.r.t.* the node feature matrix $\tilde{\boldsymbol{X}}$ and the graph structure $\tilde{\boldsymbol{A}}$,

and the lower-level optimization *w.r.t.* the couplings between current $\tilde{G}$ and $G_1, G_2$ denotes as $\boldsymbol{\pi}_1, \boldsymbol{\pi}_2$. Inspired by [28, 26, 33], we propose to solve Eq.2 using a Block Coordinate Descent algorithm, which iteratively minimizes the lower-level and the upper-level with a nested loop framework. The algorithm is presented in Alg.1. The inner loop solves the lower-level problem (i.e., FGW distance and the optimal couplings $\boldsymbol{\pi}_1^{(k)}, \boldsymbol{\pi}_2^{(k)}$) based on $\tilde{\boldsymbol{X}}^{(k)}, \tilde{\boldsymbol{A}}^{(k)}$ at the $k$-th outer loop iteration, which is non-convex and requires optimization algorithms introduced in Section 2.1. The outer loop solves the upper-level problem (i.e., minimizing the weighted sum of FGW distances), which is a convex quadratic optimization problem w.r.t. $\tilde{\boldsymbol{X}}^{(k)}$ and $\tilde{\boldsymbol{A}}^{(k)}$ with exact analytical solution (i.e., Line 7,8).

---

**Algorithm 1** `FGWMixup`: Solving Eq.2 with BCD Algorithm

---

1: **Input:** $\tilde{\boldsymbol{\mu}}, G_1 = (\boldsymbol{\mu_1}, \boldsymbol{X_1}, \boldsymbol{A_1}), G_2 = (\boldsymbol{\mu_2}, \boldsymbol{X_2}, \boldsymbol{A_2})$
2: **Optimizing:** $\tilde{\boldsymbol{X}} \in \mathbb{R}^{\tilde{n} \times d}, \tilde{\boldsymbol{A}} \in \mathbb{S}_{\tilde{n}}(\mathbb{R}), \boldsymbol{\pi}_1 \in \Pi(\tilde{\boldsymbol{\mu}}, \boldsymbol{\mu_1}), \boldsymbol{\pi}_2 \in \Pi(\tilde{\boldsymbol{\mu}}, \boldsymbol{\mu_2})$.
3: **for** $k$ in *outer iterations* and not converged **do**:
4:   $\tilde{G}^{(k)} := (\tilde{\boldsymbol{\mu}}, \tilde{\boldsymbol{X}}^{(k)}, \tilde{\boldsymbol{A}}^{(k)})$
5:   Solve $\arg\min_{\boldsymbol{\pi}_1^{(k)}} \mathrm{FGW}(\tilde{G}^{(k)}, G_1)$ with MD or CG (inner iterations)
6:   Solve $\arg\min_{\boldsymbol{\pi}_2^{(k)}} \mathrm{FGW}(\tilde{G}^{(k)}, G_2)$ with MD or CG (inner iterations)
7:   Update $\tilde{\boldsymbol{A}}^{(k+1)} \leftarrow \frac{1}{\tilde{\boldsymbol{\mu}}\tilde{\boldsymbol{\mu}}^\top}(\lambda\boldsymbol{\pi}_1^{(k)}\boldsymbol{A}_1{\boldsymbol{\pi}_1^{(k)}}^\top + (1-\lambda)\boldsymbol{\pi}_2^{(k)}\boldsymbol{A}_2{\boldsymbol{\pi}_2^{(k)}}^\top)$
8:   Update $\tilde{\boldsymbol{X}}^{(k+1)} \leftarrow \lambda\mathrm{diag}(1/\tilde{\boldsymbol{\mu}})\boldsymbol{\pi}_1^{(k)}\boldsymbol{X}_1 + (1-\lambda)\mathrm{diag}(1/\tilde{\boldsymbol{\mu}})\boldsymbol{\pi}_2^{(k)}\boldsymbol{X}_2$
9: **end for**
10: **return** $\tilde{G}^{(k)}, y_{\tilde{G}} = \lambda y_{G_1} + (1-\lambda)y_{G_2}$

---

## 2.3 Accelerating `FGWMixup`

Algorithm 1 provides a practical solution to optimize Eq.2, whereas the computation complexity is relatively high. Specifically, Alg.1 adopts a nested double-loop framework, where the inner loop is the FGW solver that optimizes the couplings $\boldsymbol{\pi}$, and the outer loop updates the optimal feature matrix $\tilde{\boldsymbol{X}}$ and graph structure $\tilde{\boldsymbol{A}}$ accordingly. However, the most common FGW solvers, such as MD and CG, require another nested double-loop algorithm. This algorithm invokes (Bregman) projected gradient descent type methods to address the non-convex optimization of FGW, which involves taking a gradient step in the outer loop and projecting to the polytope-constrained feasible set *w.r.t.* the couplings in the inner loop (e.g., using Sinkhorn [30] or Dykstra [34] iterations). Consequently, this makes the entire mixup algorithm a triple-loop framework, resulting in a heavy computation burden.

Therefore, we attempt to design a method that efficiently accelerates the algorithm. There lie two efficiency bottlenecks of Alg.1: 1) the polytope constraints of couplings makes the FGW solver a nested double-loop algorithm, 2) the strict projection of couplings to the feasible sets probably modifies the gradient step to another direction that deviates from the original navigation of the gradient, possibly leading to a slower convergence rate. In order to alleviate the problems, we are motivated to slightly relax the feasibility constraints of couplings to speed up the algorithm. Inspired by [29, 30], we do not strictly project $\boldsymbol{\pi}$ to fit the polytope constraint $\Pi(\boldsymbol{\mu_i}, \boldsymbol{\mu_j}) := \{\boldsymbol{\pi} \in \mathbb{R}_+^{n_1 \times n_2} | \boldsymbol{\pi}\mathbf{1}_{n_2} = \boldsymbol{\mu_1}, \boldsymbol{\pi}^\top\mathbf{1}_{n_1} = \boldsymbol{\mu_2}\}$ after taking each gradient step. Instead, we relax the constraint into two simplex constraints of rows and columns respectively (i.e., $\Pi_1 := \{\boldsymbol{\pi} \in \mathbb{R}_+^{n_1 \times n_2} | \boldsymbol{\pi}\mathbf{1}_{n_2} = \boldsymbol{\mu_1}\}, \Pi_2 := \{\boldsymbol{\pi} \in \mathbb{R}_+^{n_1 \times n_2} | \boldsymbol{\pi}^\top\mathbf{1}_{n_1} = \boldsymbol{\mu_2}\}$), and project $\boldsymbol{\pi}$ to the relaxed constraints $\Pi_1$ and $\Pi_2$ in an alternating fashion. The accelerated algorithm is presented in Alg.2, dubbed `FGWMixup`$_*$.

Alg.2 mainly substitutes Line 5 and Line 6 of Alg.1 with a single-loop FGW distance solver (Line 7-12 in Alg.2) that relaxes the joint polytope constraints of the couplings. Specifically, we remove the constant term in FGW distance, and denote the equivalent metric as $\overline{\mathrm{FGW}}(G_1, G_2)$. The optimization objective of $\overline{\mathrm{FGW}}(G_1, G_2)$ can be regarded as a function of $\boldsymbol{\pi}$, and we name it the FGW function $f(\boldsymbol{\pi})$ (See Appendix A.1 for details). Then we employ entropic regularization on $f(\boldsymbol{\pi})$ and select Mirror Descent as the core algorithm for the FGW solver. With the negative entropy $\phi(x) = \sum_i x_i \log x_i$ as the Bregman projection, the MD update takes the form of:

$$\boldsymbol{\pi} \leftarrow \boldsymbol{\pi} \odot \exp(-\gamma\nabla_{\boldsymbol{\pi}}f(\boldsymbol{\pi})), \quad \boldsymbol{\pi} \leftarrow \mathrm{Proj}_{\boldsymbol{\pi}\in\Pi_i}(\boldsymbol{\pi}) := \arg\min_{\boldsymbol{\pi}^*\in\Pi_i}\|\boldsymbol{\pi}^* - \boldsymbol{\pi}\|, \tag{3}$$

where $\gamma$ is the step size. The subgradient of FGW *w.r.t.* the coupling $\boldsymbol{\pi}$ can be calculated as:

$$\nabla_{\boldsymbol{\pi}}f(\boldsymbol{\pi}) = (1-\alpha)\boldsymbol{D} - 4\alpha\boldsymbol{A}_1\boldsymbol{\pi}\boldsymbol{A}_2, \tag{4}$$

**Algorithm 2** `FGWMixup*`: Accelerated FGWMixup

---

1: **Input:** $\tilde{\boldsymbol{\mu}}$, $G_1 = (\boldsymbol{\mu_1}, \boldsymbol{X_1}, \boldsymbol{A_1})$, $G_2 = (\boldsymbol{\mu_2}, \boldsymbol{X_2}, \boldsymbol{A_2})$
2: **Optimizing:** $\tilde{\boldsymbol{X}} \in \mathbb{R}^{\tilde{n} \times d}$, $\tilde{\boldsymbol{A}} \in \mathbb{S}_{\tilde{n}}(\mathbb{R})$, $\boldsymbol{\pi}_1 \in \Pi(\tilde{\boldsymbol{\mu}}, \boldsymbol{\mu_1})$, $\boldsymbol{\pi}_2 \in \Pi(\tilde{\boldsymbol{\mu}}, \boldsymbol{\mu_2})$.
3: **for** $k$ in *outer iterations* and not converged **do**:
4:      $\tilde{G}^{(k)} := (\tilde{\boldsymbol{\mu}}, \tilde{\boldsymbol{X}}^{(k)}, \tilde{\boldsymbol{A}}^{(k)})$
5:      $\boldsymbol{D}_1^{(k)} := \left( d(\tilde{\boldsymbol{X}}^{(k)}[i], \boldsymbol{X}_1[j]) \right)_{\tilde{n} \times n_1}$, $\boldsymbol{D}_2^{(k)} := \left( d(\tilde{\boldsymbol{X}}^{(k)}[i], \boldsymbol{X}_2[j]) \right)_{\tilde{n} \times n_2}$
6:      **for** $i$ in $\{1, 2\}$ **do**
7:          **while** not convergence **do**:                 ▷ Solve $\arg\min_{\boldsymbol{\pi}_i^{(k)}} \text{FGW}(\tilde{G}^{(k)}, G_i)$
8:               $\boldsymbol{\pi}_i^{(k)} \leftarrow \boldsymbol{\pi}_i^{(k)} \odot \exp\left( \gamma(4\alpha \tilde{\boldsymbol{A}}^{(k)} \boldsymbol{\pi}_i^{(k)} \boldsymbol{A}_i - (1-\alpha)\boldsymbol{D}_i^{(k)}) \right)$
9:               $\boldsymbol{\pi}_i^{(k)} \leftarrow \text{diag}(\tilde{\boldsymbol{\mu}}./\boldsymbol{\pi}_i^{(k)} \mathbf{1}_{\tilde{n}}) \boldsymbol{\pi}_1^{(k)}$         ▷ Bregman Projection on row constraint
10:             $\boldsymbol{\pi}_i^{(k)} \leftarrow \boldsymbol{\pi}_i^{(k)} \odot \exp\left( \gamma(4\alpha \tilde{\boldsymbol{A}}^{(k)} \boldsymbol{\pi}_i^{(k)} \boldsymbol{A}_i - (1-\alpha)\boldsymbol{D}_i^{(k)}) \right)$
11:             $\boldsymbol{\pi}_i^{(k)} \leftarrow \boldsymbol{\pi}_i^{(k)} \text{diag}(\boldsymbol{\mu}_i./\boldsymbol{\pi}_i^{(k)^\top} \mathbf{1}_{n_i})$     ▷ Bregman Projection on column constraint
12:          **end while**
13:      **end for**
14:      Update $\tilde{\boldsymbol{A}}^{(k+1)} \leftarrow \frac{1}{\tilde{\boldsymbol{\mu}}\tilde{\boldsymbol{\mu}}^\top}(\lambda \boldsymbol{\pi}_1^{(k)} \boldsymbol{A}_1 \boldsymbol{\pi}_1^{(k)^\top} + (1-\lambda)\boldsymbol{\pi}_2^{(k)} \boldsymbol{A}_2 \boldsymbol{\pi}_2^{(k)^\top})$
15:      Update $\tilde{\boldsymbol{X}}^{(k+1)} \leftarrow \lambda \text{diag}(1/\tilde{\boldsymbol{\mu}})\boldsymbol{\pi}_1^{(k)} \boldsymbol{X}_1 + (1-\lambda)\text{diag}(1/\tilde{\boldsymbol{\mu}})\boldsymbol{\pi}_2^{(k)} \boldsymbol{X}_2$
16: **end for**
17: **return** $\tilde{G}^{(k)}$, $y_{\tilde{G}} = \lambda y_{G_1} + (1-\lambda)y_{G_2}$

---

where $\boldsymbol{D} = (d(\boldsymbol{X}_1[i], \boldsymbol{X}_2[j]))_{n_1 \times n_2}$ is the distance matrix of node features between two graphs. The detailed derivation can be found in Appendix A.2.

Our relaxation is conducted in Line 9 and 11, where the couplings are projected to the row and column simplexes alternately instead of directly to the strict polytope. Although this relaxation may sacrifice some feasibility due to the relaxed projection, the efficiency of the algorithm has been greatly promoted. On the one hand, noted that $\Pi_1$ and $\Pi_2$ are both simplexes, the Bregman projection *w.r.t.* the negative entropy of a simplex can be extremely efficiently conducted without invoking extra iterative optimizations (i.e., Line 9, 11), which simplifies the FGW solver from a nested double-loop framework to a single-loop one. On the other hand, the relaxed constraint may also increase the convergence efficiency due to a closer optimization path to the unconstrained gradient descent.

We also provide some theoretical results to justify our algorithm. Proposition 1 presents a convergence rate analysis on our algorithm. Taking $1/\gamma$ as the entropic regularization coefficient, our FGW solver can be formulated as Sinkhorn iterations, whose convergence rate can be optimized from $\mathcal{O}(t^{-1})$ to $\mathcal{O}(t^{-2})$ by conducting marginal constraint relaxation. Proposition 2 presents a controlled gap between the optima given by the relaxed single-loop FGW solver and the strict FGW solver.

**Proposition 1.** *Let $(\mathcal{X}, \mu), (\mathcal{Y}, \nu)$ be Polish probability spaces, and $\pi \in \Pi(\mu, \nu)$ the probability measure on $\mathcal{X} \times \mathcal{Y}$ with marginals $\mu, \nu$. Let $\pi_t$ be the Sinkhorn iterations $\pi_{2t} = \arg\min_{\Pi(*, \nu)} H(\cdot | \pi_{2t-1})$, $\pi_{2t+1} = \arg\min_{\Pi(\mu, *)} H(\cdot | \pi_{2t})$, where $H(p|q) = -\sum_i p_i \log \frac{q_i}{p_i}$ is the Kullback-Leibler divergence, and $\Pi(*, \nu)$ is the set of measures with second marginal $\nu$ and arbitrary first marginal ($\Pi(\mu, *)$ is defined analogously). Let $\pi^*$ be the unique optimal solution. We have the convergence rate as follows:*

$$H(\pi_t | \pi^*) + H(\pi^* | \pi_t) = \mathcal{O}(t^{-1}), \tag{5}$$

$$H(\mu_t | \mu) + H(\mu | \mu_t) + H(\nu_t | \nu) + H(\nu | \nu_t) = \mathcal{O}(t^{-2}), \tag{6}$$

Proposition 1 implies the faster convergence of marginal constraints than the strict joint constraint. This entails that with $t$ Sinkhorn iterations of solving FGW, the solution of the relaxed FGW solver moves further than the strict one. This will benefit the convergence rate of the whole algorithm with a larger step size of $\boldsymbol{\pi}_i^{(k)}$ in each outer iteration.

**Proposition 2.** *Let $C_1, C_2$ be two convex sets, and $f(\pi)$ denote the FGW function w.r.t. $\pi$. We denote $\mathcal{X}$ as the critical point set of strict FGW that solves $\min_\pi f(\pi) + \mathbb{I}_{\pi \in C_1} + \mathbb{I}_{\pi \in C_2}$, defined by: $\mathcal{X} = \{\pi \in C_1 \cap C_2 : 0 \in \nabla f(\pi) + \mathcal{N}_{C_1}(\pi) + \mathcal{N}_{C_2}(\pi)\}$, and $\mathcal{N}_C(\pi)$ is the normal cone to $C$*

*at $\pi$. The fix-point set $\mathcal{X}_{rel}$ of the relaxed FGW solving $\min_{\pi,\omega} f(\pi) + f(\omega) + h(\pi) + h(\omega)$ is defined by: $\mathcal{X}_{rel} = \{\pi \in C_1, \omega \in C_2 : 0 \in \nabla f(\pi) + \rho(\nabla h(\omega) - \nabla h(\pi)) + \mathcal{N}_{C_1}(\pi), and\ 0 \in \nabla f(\omega) + \rho(\nabla h(\pi) - \nabla h(\omega)) + \mathcal{N}_{C_2}(\omega)\}$ where $h(\cdot)$ is the Bregman projection function. Then, the gap between $\mathcal{X}_{rel}$ and $\mathcal{X}$ satisfies:*

$$\exists \tau \in \mathbb{R},\ \forall (\pi^*, \omega^*) \in \mathcal{X}_{rel},\ \mathrm{dist}(\frac{\pi^* + \omega^*}{2}, \mathcal{X}) := \min_{x \in \mathcal{X}} \left\| \frac{\pi^* + \omega^*}{2} - x \right\| \le \tau/\rho. \tag{7}$$

This bounded gap ensures the correctness of `FGWMixup`$_*$ that as long as we select a step size $\gamma = 1/\rho$ that is small enough, the scale of the upper bound $\tau/\rho$ will be sufficiently small to ensure the convergence to the ideal optimum of our algorithm. The detailed proofs of Propositions 1 and 2 are presented in Appendix B.

## 3 Experiments

### 3.1 Experimental Settings

**Datasets**  We evaluate our methods with five widely-used graph classification tasks from the graph benchmark dataset collection TUDataset [35]: NCI1 and NCI109 [36, 37] for small molecule classification, PROTEINS [38] for protein categorization, and IMDB-B and IMDB-M [5] for social networks classification. Noted that there are no node features in IMDB-B and IMDB-M datasets, we augment the two datasets with node degree features as in [2, 19, 20]. Detailed statistics on these datasets are reported in Appendix D.1.

**Backbones**  Most existing works select traditional MPNNs such as GCN and GIN as the backbone. However, traditional MPNNs exhibit limited expressive power and sensitivity to graph structures (upper bounded by 1-Weisfeiler-Lehman (1-WL) test [17, 37]), while there exist various adavanced GNN architectures [13, 14, 16] with stronger expressive power (upper bound promoted to 3-WL test). Moreover, the use of a global pooling layer (e.g., mean pooling) in graph classification models may further deteriorate their perception of intricate graph topologies. These challenges may undermine the reliability of the conclusions regarding the effectiveness of graph structure augmentation. Therefore, we attempt to alleviate the problems from two aspects. 1) We modify the READOUT approach of GIN and GCN to save as much structural information as we can. Following [39], we apply a virtual node that connects with all other nodes and use the final latent representation of the virtual node to conduct READOUT. The two modified backbones are dubbed vGIN and vGCN. 2) We select two Transformer-based GNNs with stronger expressive power as the backbones, namely Graphormer [39] and Graphormer-GD [16]. They are proven to be more sensitive to graph structures, and Graphormer-GD is even capable of perceiving cut edges and cut vertices in graphs. Detailed information about the selected backbones is introduced in Appendix D.2.

**Comparison Baselines**  We select the following data augmentation methods as the comparison baselines. DropEdge [40] randomly removes a certain ratio of edges from the input graphs. DropNode [41] randomly removes a certain portion of nodes as well as the edge connections. M-Mixup [42] conducts Euclidean mixup in the latent spaces, which interpolates the graph representations after the READOUT function. ifMixup [19] applies an arbitrary node matching strategy to conduct mixup on graph node signals, without preserving key topologies of original graphs. $\mathcal{G}$-Mixup [20] conducts Euclidean addition on the estimated graphons of different classes of graphs to conduct class-level graph mixup. We also present the performances of the vanilla backbones, as well as our proposed method w/ and w/o acceleration, denoted as `FGWMixup`$_*$ and `FGWMixup` respectively.

**Experimental Setups**  For a fair comparison, we employ the same set of hyperparameter configurations for all data augmentation methods in each backbone architecture. For all datasets, We randomly hold out a test set comprising 10% of the entire dataset and employ 10-fold cross-validation on the remaining data. We report the average and standard deviation of the accuracy on the test set over the best models selected from the 10 folds. This setting is more realistic than reporting results from validation sets in a simple 10-fold CV and allows a better understanding of the generalizability [43]. We implement our backbones and mixup algorithms based on Deep Graph Library (DGL) [44] and Python Optimal Transport (POT) [45] open-source libraries. More experimental and implementation details are introduced in Appendix D.3 and D.4.

| | PROTEINS | | NCI1 | | NCI109 | | IMDB-B | | IMDB-M | |
|---|---|---|---|---|---|---|---|---|---|---|
| Methods | vGIN | vGCN | vGIN | vGCN | vGIN | vGCN | vGIN | vGCN | vGIN | vGCN |
| vanilla | 74.93(3.02) | 74.75(2.60) | 76.98(1.87) | 76.91(1.80) | 75.70(1.85) | 75.89(1.35) | 71.30(4.96) | 72.30(4.34) | 49.00(2.64) | 49.47(3.76) |
| DropEdge | 73.59(2.50) | 74.48(4.18) | 76.47(2.85) | 76.16(2.04) | 75.38(2.05) | 75.77(1.55) | 73.30(3.85) | 73.30(3.29) | 49.47(2.66) | 49.40(3.52) |
| DropNode | 74.48(2.91) | 75.11(3.00) | 76.89(1.25) | 77.42(1.71) | 73.98(2.16) | 75.45(1.90) | 71.50(3.23) | 73.20(5.58) | 49.80(3.29) | 50.00(3.41) |
| M-Mixup | 74.40(3.00) | 75.65(4.51) | 76.45(3.39) | 77.76(2.75) | 75.41(2.78) | 75.79(1.85) | 72.20(4.83) | 72.80(4.45) | 49.13(3.25) | 49.47(2.56) |
| ifMixup | 74.76(3.71) | 74.04(2.27) | 76.16(1.78) | 77.37(2.56) | 76.13(1.87) | 76.74(1.56) | 72.50(3.98) | 72.40(5.14) | 49.07(3.16) | 49.73(4.67) |
| $\mathcal{G}$-Mixup | 74.84(2.99) | 74.57(2.88) | 76.42(1.79) | 77.79(1.88) | 75.55(2.32) | 76.38(1.79) | 72.40(4.82) | 72.20(6.45) | 49.47(4.73) | 49.60(3.90) |
| FGWMixup | 75.02(3.86) | **76.01(3.19)** | **78.32(2.65)** | 78.37(2.40) | 76.40(1.65) | **76.79(1.81)** | 73.00(3.69) | 73.40(5.12) | **49.80(2.63)** | **50.80(4.06)** |
| FGWMixup$_*$ | **75.20(3.30)** | 75.20(3.03) | 77.27(2.71) | **78.47(1.74)** | **76.64(2.60)** | 76.52(1.59) | **73.50(4.54)** | **74.00(2.90)** | 49.20(3.38) | 50.47(5.44) |

| | Graphormer | GraphormerGD | Graphormer | GraphormerGD | Graphormer | GraphormerGD | Graphormer | GraphormerGD | Graphormer | GraphormerGD |
|---|---|---|---|---|---|---|---|---|---|---|
| Methods | | | | | | | | | | |
| vanilla | 75.47(3.16) | 76.01(2.02) | 61.56(3.70) | 77.49(2.01) | 65.54(3.04) | 74.99(1.24) | 70.40(5.00) | 71.50(4.20) | 48.87(4.10) | 47.47(2.98) |
| DropEdge | 75.20(4.02) | 75.12(3.22) | 63.07(3.21) | 74.94(2.44) | 66.73(3.50) | 74.73(3.22) | 71.10(5.65) | 72.30(3.93) | 49.60(4.09) | 46.67(3.85) |
| DropNode | 75.20(2.13) | 76.28(3.49) | 64.96(2.18) | 76.20(1.95) | 63.73(3.46) | 74.78(2.07) | 71.60(5.18) | 71.30(5.18) | 48.47(4.08) | 47.67(2.83) |
| M-Mixup | 75.11(3.78) | 74.39(3.83) | 62.31(3.48) | 75.47(1.45) | 66.54(2.70) | 74.61(1.86) | 71.10(4.83) | 70.50(4.70) | 49.67(4.25) | 48.00(3.85) |
| $\mathcal{G}$-Mixup | 75.74(3.12) | 74.85(3.52) | 63.07(4.40) | 76.06(3.12) | 65.03(2.98) | 74.90(2.04) | 72.10(6.38) | 71.10(5.01) | 46.93(5.18) | 46.80(4.41) |
| FGWMixup | **76.82(2.35)** | **77.18(3.48)** | **66.45(2.58)** | **78.20(1.88)** | 67.36(3.21) | **76.01(3.04)** | **72.60(5.08)** | **72.40(4.48)** | 49.73(3.80) | **48.87(4.03)** |
| FGWMixup$_*$ | 76.19(3.20) | 76.46(3.41) | 64.26(3.25) | 76.62(3.06) | **67.46(2.82)** | 75.45(1.80) | 71.70(4.17) | 71.90(4.35) | **50.27(4.26)** | 48.53(2.95) |

Table 1: Test set classification accuracy (%) from 10-fold CV on five benchmark datasets. Results are presented with the form of avg.(stddev.), and the best-performing method is highlighted in **boldface**. Note: ifMixup has no results on Graphormers because it generates graphs with continuous edge weights and is inapplicable for the Spatial Encoding module in the Graphormer-based architectures.

## 3.2 Experimental Results

**Main Results** We compare the performance of various GNN backbones on five benchmark datasets equipped with different graph data augmentation methods and summarize the results in Table 1. As is shown in the table, FGWMixup and FGWMixup$_*$ consistently outperform all other SOTA baseline methods, which obtain 13 and 7 best performances out of all the 20 settings respectively. The superiority is mainly attributed to the adoption of the FGW metric. Existing works hardly consider the node matching problem to align two unregistered graphs embedded in the signal-structure fused metric spaces, whereas the FGW metric searches the optimum from all possible couplings and conducts semantic-invariant augmentation guided by the optimal node matching strategy. Remarkably, despite introducing some infeasibility for acceleration, FGWMixup$_*$ maintains consistent performance due to the theoretically controlled gap with the ideal optimum, as demonstrated in Prop. 1. Moreover, FGWMixup and FGWMixup$_*$ both effectively improve the performance and generalizability of various GNNs. Specifically, our methods achieve an average relative improvement of 1.79% on MPNN backbones and 2.67% on Graphormer-based backbones when predicting held-out unseen samples. Interestingly, most SOTA graph data augmentation methods fail to bring notable improvements and even degrade the performance of Graphormers, which exhibit stronger expressive power and conduct more comprehensive interactions between graph signal and structure spaces. For instance, none of the baseline methods improve GraphormerGD on NCI109 and NCI1 datasets. However, our methods show even larger improvements on Graphormers compared to MPNNs, as we explicitly consider the interactions between the graph signal and structure spaces during the mixup process. In particular, the relative improvements of FGWMixup reach up to 7.94% on Graphormer and 2.95% on GraphormerGD. All the results above validate the effectiveness of our methods. Furthermore, we also conduct experiments on large-scale OGB [46] datasets, and the results are provided in Appendix E.5.

**Robustness against Label Corruptions** In this subsection, we evaluate the robustness of our methods against noisy labels. Practically, we introduce random label corruptions (i.e., switching to another random label) with ratios of 20%, 40%, and 60% on the IMDB-B and NCI1 datasets. We employ vGCN as the backbone model and summarize the results in Table 2. The results evidently demonstrate that the mixup-series methods consistently outperform the in-sample augmentation method (DropEdge) by a significant margin. This improvement can be attributed to the soft-labeling strategy employed by mixup, which reduces the model's sensitivity to individual mislabeled instances and encourages the model to conduct more informed predictions based on the overall distribution of the blended samples. Notably, among all the mixup methods, FGWMixup and FGWMixup$_*$ exhibit stable and consistently superior performance under noisy label conditions. In conclusion, our methods effectively enhance the robustness of GNNs against label corruptions.

**Analysis on Mixup Efficiency** We run FGWMixup and FGWMixup$_*$ individually with identical experimental settings (including stopping criteria and hyper-parameters such as mixup ratio, etc.) on the same computing device and investigate the run-time computational efficiencies. Table 3 illustrates

| Methods | IMDB-B | | | NCI1 | | |
|---|---|---|---|---|---|---|
| | 20% | 40% | 60% | 20% | 40% | 60% |
| vanilla | 70.00(5.16) | 59.70(5.06) | 47.90(4.30) | 70.58(1.29) | 61.95(2.19) | 48.25(4.87) |
| DropEdge | 68.30(5.85) | 59.40(5.00) | 50.10(1.92) | 69.51(2.27) | 60.32(2.60) | 49.61(1.28) |
| M-Mixup | 70.70(5.90) | 59.70(5.87) | 50.90(1.81) | 71.53(2.75) | 63.24(2.59) | 48.66(3.02) |
| $\mathcal{G}$-Mixup | 67.50(4.52) | 59.10(4.74) | 49.40(2.87) | 72.46(1.95) | 63.26(4.39) | 50.01(1.26) |
| FGWMixup | 70.10(4.39) | **61.90(6.17)** | 50.80(3.19) | **72.92(1.56)** | 62.99(1.35) | **50.12(3.51)** |
| FGWMixup$_*$ | **70.80(3.97)** | 61.80(5.69) | **51.00(1.54)** | 72.75(2.29) | **63.55(2.60)** | 50.02(3.38) |

Table 2: Experimental results of robustness against label corruption with different ratios.

| | Avg. Mixup Time (s) / Fold | | | | |
|---|---|---|---|---|---|
| Datasets | PROTEINS | NCI1 | NCI109 | IMDB-B | IMDB-M |
| FGWMixup | 802.24 | 1711.45 | 1747.24 | 296.62 | 212.53 |
| FGWMixup$_*$ | **394.57** | **637.41** | **608.61** | **85.69** | **74.53** |
| Speedup | 2.03× | 2.67× | 2.74× | 3.46× | 2.85× |

Table 3: Comparisons of algorithm execution efficiency between FGWMixup and FGWMixup$_*$.

the average time spent on the mixup procedures of FGWMixup and FGWMixup$_*$ per fold. We can observe that FGWMixup$_*$ decreases the mixup time cost by a distinct margin, providing at least 2.03× and up to 3.46× of efficiency promotion. The results are consistent with our theoretical analysis of the convergence rate improvements, as shown in Proposition 1. More detailed efficiency statistics and discussions are introduced in Appendix E.3.

**Infeasibility Analysis on the Single-loop FGW Solver in** FGWMixup$_*$   We conduct an experiment to analyze the infeasibility of our single-loop FGW solver compared with the strict CG solver. Practically, we randomly select 1,000 pairs of graphs from PROTEINS dataset and apply the two solvers to calculate the FGW distance between each pair of graphs. The distances of the $i$-th pair of graphs calculated by the strict solver and the relaxed solver are denoted as $d_i$ and $d_i^*$, respectively. We report the following metrics for comparison:

- **MAE**: Mean absolute error of FGW distance, i.e., $\frac{1}{N}\sum |d_i - d_i^*|$.
- **MAPE**: Mean absolute percentage error of FGW distance, i.e., $\frac{1}{N}\sum \frac{|d_i - d_i^*|}{d_i}$.
- **mean-FGW**: Mean FGW distance given by the strict CG solver, i.e., $\frac{1}{N}\sum d_i$.
- **mean-FGW\***: Mean FGW distance given by the single-loop solver, i.e., $\frac{1}{N}\sum d_i^*$.
- **T-diff**: L2-norm of the difference between two transportation plan matrices (divided by the size of the matrix for normalization).

| MAE | MAPE | mean-FGW | mean-FGW* | T-diff |
|---|---|---|---|---|
| 0.0126(0.0170) | 0.0748(0.1022) | 0.2198 | 0.2143 | 0.0006(0.0010) |

Table 4: Infeasibility analysis on the single-loop FGW solver in FGWMixup$_*$.

The results are shown in Table 4. We can observe that the MAPE is only 0.0748, which means the FGW distance estimated by the single-loop relaxed solver is only 7.48% different from the strict CG solver. Moreover, the absolute error is around 0.01, which is quite small compared with the absolute value of FGW distances (~0.21). We can also find that the L2-norm of the difference between two transportation plan matrices is only 0.0006, which means two solvers give quite similar transportation plans. All the results imply that the single-loop solver will not produce huge infeasibility or make the estimation of FGW distance inaccurate.

### 3.3   Further Analyses

**Effects of the Trade-off Coefficient** $\alpha$   We provide sensitivity analysis w.r.t. the trade-off coefficient $\alpha$ of our proposed mixup methods valued from {0.05, 0.5, 0.95, 1.0} on two backbones and two

| Methods | PROTEINS | | | | NCI1 | | | |
|---|---|---|---|---|---|---|---|---|
| | $\alpha$=0.95 | $\alpha$=0.5 | $\alpha$=0.05 | $\alpha$=1.0 | $\alpha$=0.95 | $\alpha$=0.5 | $\alpha$=0.05 | $\alpha$=1.0 |
| vGIN-FGWMixup | 75.02(3.86) | **75.38(2.58)** | 74.86(2.40) | 74.57(2.62) | **78.32(2.65)** | 77.42(1.93) | 77.62(2.37) | 75.91(2.93) |
| vGCN-FGWMixup | **76.01(3.19)** | 75.47(3.56) | 74.93(2.74) | 74.40(3.57) | **78.37(2.40)** | 77.93(1.68) | 78.00(1.00) | 77.27(0.92) |
| vGIN-FGWMixup$_*$ | **75.20(3.30)** | 74.57(3.30) | 74.39(3.01) | 73.94(3.86) | **77.27(2.71)** | 77.23(2.47) | 76.59(2.14) | 77.20(1.69) |
| vGCN-FGWMixup$_*$ | **75.20(3.03)** | 74.57(3.52) | 74.84(3.16) | 74.66(2.91) | 78.47(1.74) | 77.66(1.48) | **78.93(1.91)** | 77.71(1.97) |

Table 5: Experimental results of different $\alpha$ on PROTEINS and NCI1 datasets.

Figure 1: Test performance of our methods with different graph size settings on NCI1 and PROTEINS datasets using vGCNs as the backbone.

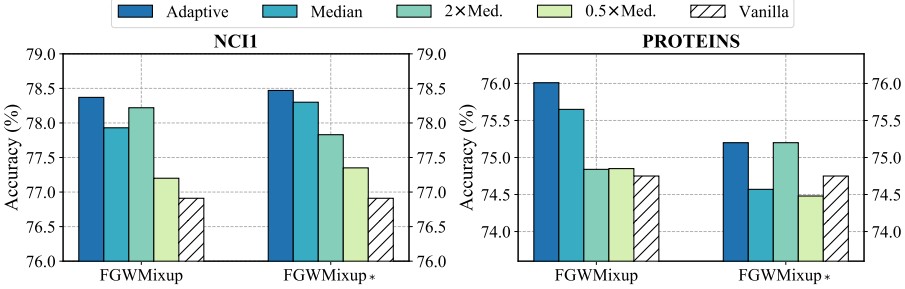

datasets. Note that $\alpha$ controls the weights of the node feature alignment and graph structure alignment costs, and $\alpha = 1.0$ falls back to the case of GW metric where node features are not incorporated. From the results shown in Table 5, we can observe that: 1) when FGW falls back to GW ($\alpha$ =1), where node features are no longer taken into account, the performance will significantly decay (generally the worst among all investigated $\alpha$ values). This demonstrates the importance of solving the joint modeling problem in graph mixup tasks. 2) $\alpha$=0.95 is the best setting in most cases. This empirically implies that it is better to conduct more structural alignment in graph mixup. In practice, we set $\alpha$ to 0.95 for all of our reported results.

**Effects of Mixup Graph Sizes** We investigate the performance of `FGWMixup` and `FGWMixup`$_*$ with various mixup graph sizes (node numbers), including adaptive graph sizes (i.e., weighted average size of the mixup source graphs, $\tilde{n} = \lambda n_1 + (1 - \lambda)n_2$, which is selected as our method) and fixed graph sizes (i.e., the median size of all training graphs, $0.5 \times$ and $2 \times$ the median). The ablation studies are composed on NCI1 and PROTEINS datasets using vGCN backbone as motivating examples, and the results are illustrated in Fig.1. We can observe that the best performance is steadily obtained when selecting graph sizes with an adaptive strategy, whereas the fixed graph sizes lead to unstable results. Specifically, selecting the median or larger size may occasionally yield comparable performances, yet selecting the smaller size can result in an overall performance decay of over 1%. This phenomenon is associated with the graph size generalization problem [47, 48] that describes the performance degradation of GNNs caused by the graph size distributional shift between training and testing data. The fixed strategy may aggravate this problem, particularly for small graphs that struggle to generalize to larger ones. In contrast, the adaptive strategy can potentially combat this distributional shift by increasing the data diversity and reach better test time performance.

**Effects of GNN Depths** To validate the improvement of our methods across various model depths, we evaluate the performance of `FGWMixup` and `FGWMixup`$_*$ using vGCNs equipped with different numbers (3-8) of layers. We experiment on NCI1 and PROTEINS datasets, and the results are illustrated in Fig.2. We can observe that our methods consistently improve the performance on NCI1 under all GCN depth settings by a significant margin. The same conclusion is also true for most cases on PROTEINS except for the 7-layer vGCN. These results indicate a universal improvement of our methods on GNNs with various depths.

**Other Discussions** More further analyses are introduced in the Appendix, including qualitative analyses of our mixup results (see Appendix E.1), further discussions on $\mathcal{G}$-Mixup (see Appendix E.2), and sensitivity analysis of the hyperparameter $k$ in Beta distribution where mixup weights are sampled from (see Appendix E.4).

Figure 2: Test performance of our methods using vGCNs with varying numbers of layers on NCI1 and PROTEINS datasets.

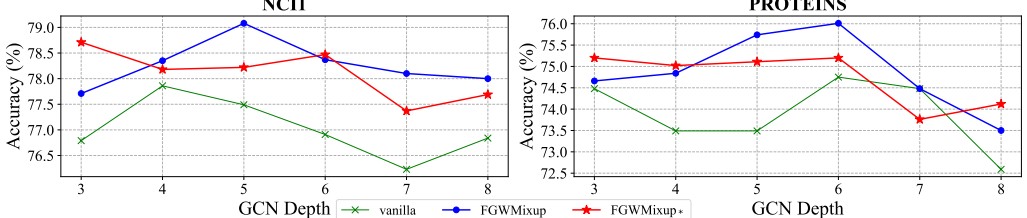

# 4 Related Works

**Graph Data Augmentation**   There are currently two mainstream perspectives of graph data augmentation for graph-level classifications. A line of research concentrates on graph signal augmentation [41, 18, 19]. Node Atrribute Masking [41] assigns random node features for a certain ratio of graph nodes. IfMixup [19] conducts Euclidean mixup on node features from different graphs with arbitrary node alignment strategy, whereas it damages the critical topologies (e.g., rings, bipartibility) of the source graphs. Another line of work focuses on graph structure augmentation [49, 20, 21, 22, 50, 41, 51]. Approaches like Subgraph[49, 41], DropEdge[40] and GAug[50] conduct removals or additions of graph nodes or edges to generate new graph structures. Graph Transplant[21] and Submix[22] realize CutMix-like augmentations on graphs, which cut off a subgraph from the original graph and replace it with another. $\mathcal{G}$-Mixup [20] estimates a graph structure generator (i.e., graphon) for each class of graphs and conducts Euclidean addition on graphons to realize structural mixup. However, as GNNs can capture the complex interaction between the two entangled yet complementary input spaces, it is essential to model this interaction for a comprehensive graph data augmentation. Regrettably, current works have devoted little attention to this interaction.

**Optimal Transport on Graphs**   Building upon traditional Optimal Transport (OT) [25] methods (e.g., Wasserstein distance[52]), graph OT allows defining a very general distance metric between the structured/relational data points embedded in different metric spaces, where the data points are modeled as probability distributions. It proceeds by solving a coupling between the distributions that minimizes a specific cost. The solution of graph OT hugely relies on the (Fused) Gromov-Wasserstein (GW) [27, 26] distance metric. Further works employ the GW couplings for solving tasks such as graph node matching and partitioning [53, 32, 54], and utilize GW distance as a common metric in graph metric learning frameworks [55, 31, 56]. Due to the high complexity of the GW-series algorithm, another line of research [28, 57, 58, 29] concentrates on boosting the computational efficiency. In our work, we formulate graph mixup as an FGW-based OT problem that solves the optimal node matching strategy minimizing the alignment costs of graph structures and signals. Meanwhile, we attempt to accelerate the mixup procedure with a faster convergence rate.

# 5 Conclusion and Limitation

In this work, we introduce a novel graph data augmentation method for graph-level classifications dubbed `FGWMixup`. `FGWMixup` formulates the mixup of two graphs as an optimal transport problem aiming to seek a "midpoint" of two graphs embedded in the graph signal-structure fused metric space. We employ the FGW distance metric to solve the problem and further propose an accelerated algorithm that improves the convergence rate from $\mathcal{O}(t^{-1})$ to $\mathcal{O}(t^{-2})$ for better scalability of our method. Comprehensive experiments demonstrate the effectiveness of `FGWMixup` in terms of enhancing the performance, generalizability, and robustness of GNNs, and also validate the efficiency and correctness of our acceleration strategy.

Despite the promising results obtained in our work, it is important to acknowledge its limitations. Our focus has primarily been on graph data augmentation for graph-level classifications. However, it still remains a challenging question to better exploit the interactions between the graph signal and structure spaces for data augmentation in other graph prediction tasks, such as link prediction and node classification. Moreover, we experiment with four classic (MPNNs) and advanced (Graphormers) GNNs, while there remain other frameworks that could be taken into account. We expect to carry out a graph classification benchmark with more comprehensive GNN frameworks in our future work.

## Acknowledgements

This work was supported by the National Natural Science Foundation of China (No.82241052).

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

# A  Problem Properties

## A.1  FGW Distance Metric

Fused Gromov-Wasserstein Distance [26] are defined in Eq.1. It can also be rewritten as follows:

$$\text{FGW}(G_1, G_2) = \min_{\boldsymbol{\pi} \in \Pi(\boldsymbol{\mu}_1, \boldsymbol{\mu}_2)} \langle (1-\alpha)\boldsymbol{D} + \alpha L(\boldsymbol{A}_1, \boldsymbol{A}_2) \otimes \boldsymbol{\pi}, \boldsymbol{\pi} \rangle, \tag{8}$$

where $\boldsymbol{D} = \{d(\boldsymbol{x}_1^{(i)}, \boldsymbol{x}_2^{(j)})\}_{ij}$ is the distance matrix of node features, and $L(\boldsymbol{A}_1, \boldsymbol{A}_2) = \{|\boldsymbol{A_1}(i,k) - \boldsymbol{A_2}(j,\ell)|\}_{i,j,k,\ell}$ is a 4-dimensional tensor that depicts the structural distances. We denote $\langle \boldsymbol{U}, \boldsymbol{V} \rangle = \text{tr}(\boldsymbol{U}^\top \boldsymbol{V})$ as the matrix scalar product. The $\otimes$ operator conducts tensor-matrix multiplication as: $\mathcal{L} \otimes T \stackrel{\text{def.}}{=} \left( \sum_{k,\ell} \mathcal{L}_{i,j,k,\ell} T_{k,\ell} \right)_{i,j}$. Then according to Proposition 1 in [28], when we take the $\ell_2$-norm to calculate the structural distance, we have:

$$L(\boldsymbol{A}_1, \boldsymbol{A}_2) \otimes \boldsymbol{\pi} = c_{\boldsymbol{A}_1, \boldsymbol{A}_2} - 2\boldsymbol{A}_1 \boldsymbol{\pi} \boldsymbol{A}_2, \tag{9}$$

where $c_{\boldsymbol{A}_1, \boldsymbol{A}_2} = \boldsymbol{A}_1^{\odot 2} \boldsymbol{\mu}_1 \mathbf{1}_{n_2}^\top + \mathbf{1}_{n_1} \boldsymbol{\mu}_2^\top \boldsymbol{A}_2^{\odot 2}$, and $\odot 2$ denotes the Hadamard square (i.e., $\boldsymbol{U}^{\odot 2} = \boldsymbol{U} \odot \boldsymbol{U}$). Taking Eq.9 into Eq.8, we can rewrite the minimizer as:

$$\begin{aligned}
\text{RHS} &= (1-\alpha)\langle \boldsymbol{D}, \boldsymbol{\pi} \rangle + \overbrace{\alpha\langle \boldsymbol{A}_1^{\odot 2} \boldsymbol{\mu}_1 \mathbf{1}_{n_2}^\top, \boldsymbol{\pi} \rangle + \alpha\langle \mathbf{1}_{n_1} \boldsymbol{\mu}_2^\top \boldsymbol{A}_2^{\odot 2}, \boldsymbol{\pi} \rangle} - 2\alpha\langle \boldsymbol{A}_1 \boldsymbol{\pi} \boldsymbol{A}_2, \boldsymbol{\pi} \rangle \\
&= \underbrace{\alpha(\boldsymbol{A}_1^{\odot 2} \boldsymbol{\mu}_1^\top \boldsymbol{\mu}_1 + \boldsymbol{A}_2^{\odot 2} \boldsymbol{\mu}_2^\top \boldsymbol{\mu}_2)}_{\text{constant}} + \langle (1-\alpha)\boldsymbol{D} - 2\alpha \boldsymbol{A}_1 \boldsymbol{\pi} \boldsymbol{A}_2, \boldsymbol{\pi} \rangle
\end{aligned} \tag{10}$$

The items in the overbrace become a constant because the marginals of $\boldsymbol{\pi}$ are fixed as $\boldsymbol{\mu}_1$ and $\boldsymbol{\mu}_2$, respectively. Therefore, we can remove the constant term and formulate an equivalent optimization objective denoted as $\overline{\text{FGW}}(G_1, G_2)$, which can be rewritten as:

$$\begin{aligned}
\overline{\text{FGW}}(G_1, G_2) &= \min_{\boldsymbol{\pi} \in \Pi(\boldsymbol{\mu}_1, \boldsymbol{\mu}_2)} \langle (1-\alpha)\boldsymbol{D} - 2\alpha \boldsymbol{A}_1 \boldsymbol{\pi} \boldsymbol{A}_2, \boldsymbol{\pi} \rangle \\
&= \min_{\boldsymbol{\pi} \in \Pi(\boldsymbol{\mu}_1, \boldsymbol{\mu}_2)} (1-\alpha) \text{tr}(\boldsymbol{\pi}^\top \boldsymbol{D}) - 2\alpha \text{tr}(\boldsymbol{\pi}^\top \boldsymbol{A}_1 \boldsymbol{\pi} \boldsymbol{A}_2).
\end{aligned} \tag{11}$$

We denote the above optimization objective as a function $f(\boldsymbol{\pi})$ w.r.t. $\boldsymbol{\pi}$, and we call it the FGW function in the main text.

## A.2  Mirror Descent and the Relaxed FGW Solver in `FGWMixup`$_*$

**Definition 1** (Normal Cone). *Given any set $\mathcal{C}$ and point $x \in \mathcal{C}$, we can define **normal cone** as:*

$$\mathcal{N}_{\mathcal{C}}(x) = \{g : g^\top x \geq g^\top y, \forall y \in \mathcal{C}\}$$

There are some properties of normal cone. The normal cone is always convex, and $\text{Proj}_{\mathcal{C}}(x + z) = x, \forall x \in \mathcal{C}, z \in \mathcal{N}_{\mathcal{C}}(x)$ always holds.

**Proposition 3** (The Update of Mirror Descent). *The update of Mirror Descent takes the form of* $x^{(k+1)} = \arg\min_{x \in \mathcal{X}} D_\phi(x, y^{(k+1)}))$, *where* $\phi(\cdot)$ *is a Bregman Projection, and* $\nabla\phi(y^{(k+1)}) = \nabla\phi(\omega^{(k)}) - \nabla f(\pi)/\rho$.

*Proof.* For a constrained optimization problem $\min_{x \in \mathcal{X}} f(x), \mathcal{X} \subset \mathcal{D}$, Mirror Descent iteratively updates $x$ with $x^{(k+1)} = \arg\min_{x \in \mathcal{X}} \nabla f(x^{(k)})^\top x + \rho D_\phi(x, x^{(k)})$ with Bregman divergence $D_\phi(x, y) := \phi(x) - \phi(y) - \langle \nabla\phi(y), x - y \rangle$. We further have:

$$\begin{aligned}
& x^{(k+1)} = \arg\min_{x \in \mathcal{X}} \nabla f(x^{(k)})^\top x + \rho D_\phi(x, x^{(k)}) \\
\Rightarrow & 0 \in \nabla f(x^{(k)})^\top / \rho + \nabla\phi(x^{(k+1)}) - \nabla\phi(x^{(k)}) + \mathcal{N}_{\mathcal{X}}(x^{(k+1)}) \\
\Rightarrow & x^{(k+1)} = \arg\min_{x \in \mathcal{X}} D_\phi(x, y^{(k+1)})), \text{ where } \nabla\phi(y^{(k+1)}) = \nabla\phi(x^{(k)}) - \nabla f(x^{(k)})/\rho
\end{aligned} \tag{12}$$

$\square$

Figure 3: The illustration of the MD procedure.

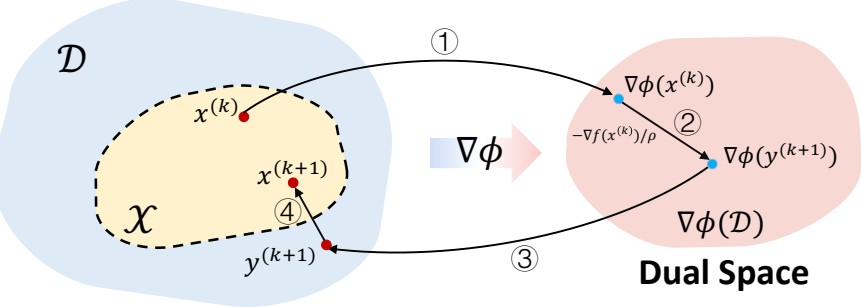

This updating procedure is vividly depicted by Fig.3. For each update of $x^{(k)}$, MD conducts the following steps:

① Apply mirror mapping ($\nabla\phi(\cdot)$) on $x^{(k)}$ from the primal space $\mathcal{D}$ to the dual space $\nabla\phi(\mathcal{D})$.

② Take a gradient step in the dual space, which is $\delta^{(k+1)} = \nabla\phi(x^{(k)}) - \nabla f(x^{(k)})/\rho$.

③ Inversely mapping ($\nabla\phi^{-1}(\cdot)$) the dual space back to the primal space, which is $y^{(k+1)} = \nabla\phi^{-1}(\delta^{(k+1)})$, also written as $\nabla\phi(y^{(k+1)}) = \nabla\phi(x^{(k)}) - \nabla f(x^{(k)})/\rho$.

④ The updated point may fall out of the constraints of $\mathcal{X}$. Thus, Bregman projection is conducted to project $y^{(k+1)}$ to $x^{(k+1)} \in \mathcal{X}$, where $x^{(k+1)} = \arg\min_{x\in\mathcal{X}} D_\phi(x, y^{(k+1)}))$.

The relaxed FGW conducts projections to constraints of rows and columns alternately, which is written as $\min_\pi f(\pi) + \mathbb{I}_{\{\pi\in\Pi_1\}} + \mathbb{I}_{\{\pi\in\Pi_2\}}$, where $\mathbb{I}$ is the indicator function. We adopt the operator splitting strategy, and the optimization can be formulated as:

$$F_\rho(\pi, \omega) = f(\pi) + f(\omega) + \mathbb{I}_{\{\pi\in\Pi_1\}} + \mathbb{I}_{\{\omega\in\Pi_2\}} \tag{13}$$

We apply Mirror Descent algorithm with Bregman divergence $D_\phi(x, y)$ to solve this problem, whose update takes the form of:

$$
\begin{aligned}
\pi^{(k+1)} &= \arg\min_{\pi\in\Pi_1} \nabla f(\pi)^\top \pi + \rho D_\phi\left(\pi, \omega^{(k)}\right) \\
\omega^{(k+1)} &= \arg\min_{\omega\in\Pi_2} \nabla f(\omega)^\top \omega + \rho D_\phi\left(\omega, \pi^{(k+1)}\right)
\end{aligned}
\tag{14}
$$

Let's take the MD update of $\pi$ as an example. Taking relative entropy as $\phi(x) = \sum_i x_i \log x_i$, we can solve $y^{(k+1)}$ (conducting steps ① to ③) with exact analytical solutions. Specifically, $\nabla\phi(x) = \sum_i (1 + \log x_i)$, and the update w.r.t. $\pi$ writes:

$$
\begin{aligned}
\pi^{(k+1)} &= \arg\min_{\pi\in\Pi_1} D_\phi(\pi, y^{(k+1)}), \text{ where } \log(y^{(k+1)}) = \log(\omega^{(k)}) - \nabla f(\omega^{(k)})/\rho \\
\Rightarrow \pi^{(k+1)} &= \arg\min_{\pi\in\Pi_1} D_\phi(\pi, y^{(k+1)}), \text{ where } y^{(k+1)} = \omega^{(k)} \odot \exp(-\nabla f(\omega^{(k)})/\rho).
\end{aligned}
\tag{15}
$$

Furthermore, under this setting, we also have an exact solution of the projection step ④ $\arg\min_{\pi\in\Pi_1} D_\phi(\pi, y^{(k+1)})) = \mathrm{diag}(\mu_1./y^{(k+1)}1_{n_2})y^{(k+1)}$. Then the MD update procedure can be formulated according to Prop. 3 as follows:

$$\pi^{(k+1)} \leftarrow \omega^{(k)} \odot \exp(-\nabla f(\omega^{(k)})/\rho), \pi^{(k+1)} \leftarrow \mathrm{diag}(\mu_1./\pi^{(k+1)}1_{n_2})\pi^{(k+1)} \tag{16}$$

Furthermore, the sub-gradient of our FGW optimizing objective *w.r.t.* $\boldsymbol{\pi}$ is calculated as follows:

$$\nabla f(\pi) = \nabla_\pi \mathrm{FGW}(G_1, G_2) = (1 - \alpha)\boldsymbol{D} - 4\alpha\boldsymbol{A}_1\pi\boldsymbol{A}_2. \tag{17}$$

Taking Eq.17 into Eq.16, we have the update of $\pi$, and a similar procedure is conducted for $\omega$. Setting $\rho$ to $1/\gamma$, the final update procedure is shown in Line 8-11 in Alg.2.

# B Proofs of Theoretical Results

## B.1 Proof of Proposition 1

*Proof.* From Corollary 4.6 of [59] we have: for any Sinkhorn iteration optimizing $\pi \in \Pi(\mu, \nu)$ on two different marginals alternately $\mu, \nu$, let $t_1 = \inf\{t \geq 0 : H(\mu_{2t}|\mu) \leq 1\} - 1$, when iterations $t \geq 2t_1$:

$$H(\mu_{2t} \mid \mu) + H(\mu \mid \mu_{2t}) \leq 10 \frac{C_1^2 \vee H(\pi_* \mid R)}{(\lfloor t/2 \rfloor - t_1) t} = \mathcal{O}(t^{-2}),$$

$$H(\pi_{2t} \mid \pi_*) + H(\pi_* \mid \pi_{2t}) \leq 5 \frac{C_1^2 \vee \left(H(\pi_* \mid R)^{1/2} C_1\right)}{\sqrt{(\lfloor t/2 \rfloor - t_1) t}} = \mathcal{O}(t^{-1}),$$

(18)

where $C_1$ is a constant, and $R$ is a fixed reference measure. $\square$

## B.2 Proof of Proposition 2

**Definition 2** (Bounded Linear Regularity, Definition 5.6 in [60]). *Let $C_1, C_2, \cdots, C_N$ be closed convex subsets of $\mathbb{R}^n$ with a non-empty intersection $C$. We call the set $\{C_1, C_2, \cdots, C_N\}$ is **bounded linearly regular** if for every bounded subset $\mathbb{B}$ of $\mathbb{R}^n$, there exists a constant $\kappa > 0$ such that*

$$d(x, C) \leq \kappa \max_{i \in \{1, \ldots, N\}} d(x, C_i), \forall x \in \mathbb{B}, \text{ where } d(x, \mathcal{C}) := \min_{x' \in \mathcal{C}} \|x - x'\|.$$

The BLR condition naturally holds if all the $C_i$ are polyhedral sets.

**Definition 3** (Strong Convexity). *A differentiable function $f(\cdot)$ is **strongly convex** with the constant $m$, if the following inequality holds for all points $x, y$ in its domain:*

$$(\nabla f(x) - \nabla f(y))^\top (x - y) \geq m \|x - y\|^2,$$

*where $\|\cdot\|$ is the corresponding norm.*

In Mirror Descent algorithm, the Bregman divergence $D_h(x, y)$ requires $h$ to be strongly convex.

*Proof.* Recall that the fixed point set of relaxed FGW is $\mathcal{X}_{rel} = \{\pi^* \in C_1, \omega^* \in C_2 : 0 \in \nabla f(\pi^*) + \rho(\nabla h(\omega^*) - \nabla h(\pi^*)) + \mathcal{N}_{C_1}(\pi^*), \text{ and } 0 \in \nabla f(\omega^*) + \rho(\nabla h(\pi^*) - \nabla h(\omega^*)) + \mathcal{N}_{C_2}(\omega^*)\}$. Let $p \in \mathcal{N}_{C_1}(\pi^*)$ and $q \in \mathcal{N}_{C_2}(\omega^*)$, $\mathcal{X}_{rel}$ is denoted as:

$$\mathcal{X}_{rel} = \{\pi^* \in C_1, \omega^* \in C_2 : \nabla f(\pi^*) + \rho(\nabla h(\omega^*) - \nabla h(\pi^*)) + p = 0, p \in \mathcal{N}_{C_1}(\pi^*) \\ \nabla f(\omega^*) + \rho(\nabla h(\pi^*) - \nabla h(\omega^*)) + q = 0, q \in \mathcal{N}_{C_2}(\omega^*)\}$$

(19)

We denote $\hat{\pi} = \text{Proj}_{C_1 \cap C_2}(\pi^*)$. According to Definition 2, as $C_1$ and $C_2$ are both polyhedral constraints and satisfy the BLR condition, we have:

$$\|\hat{\pi} - \pi^*\| + \|\hat{\pi} - \omega^*\| \leq 2 \|\hat{\pi} - \pi^*\| + \|\pi^* - \omega^*\| \\ = 2d(\pi^*, C_1 \cap C_2) + \|\pi^* - \omega^*\| \\ \leq (2\kappa + 1) \|\pi^* - \omega^*\|.$$

(20)

The first inequality holds based on the triangle inequality of the norm distances, and the second inequality holds based on the definition of point-to-space distance in Def.2. Moreover, according to the definition of $\mathcal{X}_{rel}$, for any $(\pi^*, \omega^*) \in \mathcal{X}_{rel}$, we have:

$$\nabla f(\omega^*)^\top (\hat{\pi} - \pi^*) + \rho(\nabla h(\pi^*) - \nabla h(\omega^*))^\top (\hat{\pi} - \pi^*) + p^\top (\hat{\pi} - \pi^*) = 0, p \in \mathcal{N}_{C_1}(\pi^*)$$

$$\nabla f(\pi^*)^\top (\hat{\pi} - \omega^*) + \rho(\nabla h(\omega^*) - \nabla h(\pi^*))^\top (\hat{\pi} - \omega^*) + q^\top (\hat{\pi} - \omega^*) = 0, q \in \mathcal{N}_{C_2}(\omega^*)$$

(21)

Summing up the equations above, we have:

$$\nabla f(\omega^*)^\top (\hat{\pi} - \pi^*) + \nabla f(\pi^*)^\top (\hat{\pi} - \omega^*) + \rho(\nabla h(\pi^*) - \nabla h(\omega^*))^\top (\omega^* - \pi^*) + p^\top (\hat{\pi} - \pi^*) + q^\top (\hat{\pi} - \omega^*)$$

$$\overset{(a)}{=} \nabla f(\omega^*)^\top (\hat{\pi} - \pi^*) + \nabla f(\pi^*)^\top (\hat{\pi} - \omega^*) - \rho(D_h(\pi^*, \omega^*) + D_h(\omega^*, \pi^*)) + p^\top (\hat{\pi} - \pi^*) + q^\top (\hat{\pi} - \omega^*) = 0$$

(22)

The $(a)$ equation holds based on the definition of Bregman Divergence $D_h(x,y) := h(x) - h(y) - \langle \nabla h(y), x-y \rangle$. Furthermore, due to the property of normal cones $\mathcal{N}_{C_1}(\pi^*), \mathcal{N}_{C_2}(\omega^*)$, we have $p^\top(\pi - \pi^*) \leq 0, \forall \pi \in C_1$ and $q^\top(\omega - \omega^*) \leq 0, \forall \omega \in C_2$. Taking $\pi$ and $\omega$ as $\hat{\pi} \in C_1 \cap C_2$, we have $p^\top(\hat{\pi} - \pi^*) + q^\top(\hat{\pi} - \omega^*) \leq 0$. Therefore, from 22 we have the following inequality:

$$
\begin{aligned}
\nabla f(\omega^*)^\top(\hat{\pi} - \pi^*) &+ \nabla f(\pi^*)^\top(\hat{\pi} - \omega^*) - \rho(D_h(\pi^*,\omega^*) + D_h(\omega^*,\pi^*)) \geq 0 \\
\Rightarrow \rho(D_h(\pi^*,\omega^*) + D_h(\omega^*,\pi^*)) &\leq \nabla f(\omega^*)^\top(\hat{\pi}-\pi^*) + \nabla f(\pi^*)^\top(\hat{\pi}-\omega^*) \\
&\leq \|f(\omega^*)\|\|\hat{\pi}-\pi^*\| + \|\nabla f(\pi^*)\|\|\hat{\pi}-\omega^*\| \\
&\overset{(b)}{\leq} C_f(\|\hat{\pi}-\pi^*\| + \|\hat{\pi}-\omega^*\|) \\
&\overset{(20)}{\leq} C_f(2\kappa+1)\|\pi^*-\omega^*\|.
\end{aligned}
\tag{23}
$$

The $(b)$ inequality holds as the optimization objective is a quadratic function with bounded constraints, which means the gradient norm has a natural constant upper bound $C_f$. For the left hand side of the inequality above, the sum of Bregman divergences is also lower bounded by

$$
D_h(\pi^*,\omega^*) + D_h(\omega^*,\pi^*) = (\nabla h(\pi^*) - \nabla h(\omega^*))^\top(\pi^*-\omega^*) \geq \sigma\|\pi^*-\omega^*\|^2, \tag{24}
$$

as $h(\cdot)$ is $\sigma$-strongly convex in the definition of Bregman Divergence. Therefore, combining (24) and (23), we have:

$$
\|\pi^*-\omega^*\| \leq \frac{C_f(2\kappa+1)}{\sigma\rho} \tag{25}
$$

Based on this important result, we analyze the distance between $\mathcal{X}$ and $\mathcal{X}_{rel}$, which is given by $\text{dist}(\frac{\pi^*+\omega^*}{2}, \mathcal{X}) := \min_{x \in \mathcal{X}} \left\| \frac{\pi^*+\omega^*}{2} - x \right\|, \pi^* \in C_1, \omega^* \in C_2$. By adding up the two equations in 19 depicting the conditions on $\pi^*$ and $\omega^*$ of $\mathcal{X}_{rel}$, we have:

$$
\nabla f(\pi^*) + \nabla f(\omega^*) + p + q = 0 \overset{(c)}{\Rightarrow} \nabla f(\frac{\pi^*+\omega^*}{2}) + \frac{p+q}{2} = 0 \tag{26}
$$

The $(c)$ derivation holds due to the linear property of the subgradient of FGW w.r.t. $\pi$ (see Eq.17). Invoking this equality, we present the following uppper bound analysis on $\text{dist}(\frac{\pi^*+\omega^*}{2}, \mathcal{X})$:

$$
\begin{aligned}
\text{dist}\left(\frac{\pi^*+w^*}{2}, \mathcal{X}\right) &\overset{(d)}{\leq} \epsilon \left\| \frac{\pi^*+w^*}{2} - \text{proj}_{C_1 \cap C_2}\left(\frac{\pi^*+w^*}{2} - \nabla f\left(\frac{\pi^*+w^*}{2}\right)\right) \right\| \\
&\overset{(e)}{=} \epsilon \left\| \frac{\pi^*+w^*}{2} - \text{proj}_{C_1 \cap C_2}\left(\frac{\pi^*+p+w^*+q}{2}\right) \right\| \\
&\overset{(f)}{=} \frac{\epsilon}{2} \left\| \text{proj}_{C_1}(\pi^*+p) + \text{proj}_{C_1}(w^*+q) - 2\,\text{proj}_{C_1 \cap C_2}\left(\frac{\pi^*+p+w^*+q}{2}\right) \right\| \\
&\overset{(g)}{\leq} \frac{M\epsilon}{2} \|\pi^*-w^*\|,
\end{aligned}
\tag{27}
$$

The $(d)$ inequality comes from the Luo-Tseng error bound condition [61] and Prop.3.1 in [29]. The $(e)$ holds based on Eq.26. The $(f)$ equality holds based on the property of the normal cone, which is $\text{Proj}_C(x+z) = x, \forall x \in C, z \in \mathcal{N}_C(x)$. The $(g)$ inequality holds by invoking Lemma 3.2 in [29], which tells $\left\| \text{proj}_{C_1}(x) + \text{proj}_{C_2}(y) - 2\,\text{proj}_{C_1 \cap C_2}\left(\frac{x+y}{2}\right) \right\| \leqslant M \left\| \text{proj}_{C_1}(x) - \text{proj}_{C_2}(y) \right\|$. Finally, combining (25) and (27), we can obtain the desired result, i.e.,

$$
\forall(\pi^*,\omega^*) \in \mathcal{X}_{rel}, \ \text{dist}(\frac{\pi^*+\omega^*}{2}, \mathcal{X}) \leq \frac{(2\kappa+1)\epsilon C_f M}{2\sigma\rho} = \tau/\rho \tag{28}
$$

where $\tau := \frac{(2\kappa+1)\epsilon C_f M}{2\sigma}$ is a constant. $\qquad\qquad\qquad\qquad\qquad\qquad\qquad\square$

## C  Complete Mixup Procedure

We provide the complete mixup procedure of our method in Algorithm 3. We conduct augmentation only on the training set, and we mixup graphs from every pair of classes. For a specific mixup of two

| Datasets | graphs | avg nodes | med nodes | avg edges | med edges | feat dim | classes |
|---|---|---|---|---|---|---|---|
| PROTEINS | 1113 | 39.05 | 26 | 72.82 | 98 | 3 | 2 |
| NCI1 | 4110 | 29.76 | 27 | 32.30 | 58 | 37 | 2 |
| NCI109 | 4127 | 29.57 | 26 | 32.13 | 58 | 38 | 2 |
| IMDB-B | 1000 | 19.77 | 17 | 193.06 | 260 | N/A | 2 |
| IMDB-M | 1500 | 13.00 | 10 | 131.87 | 144 | N/A | 3 |

Table 6: Detailed statistics of experimented datasets.

graphs, we apply our proposed algorithm $\texttt{FGWMixup}_{(*)}$ (Alg.1, 2) to obtain the synthetic mixup graph $\tilde{G} = (\tilde{\boldsymbol{\mu}}, \tilde{\boldsymbol{X}}, \tilde{\boldsymbol{A}})$. However, the numerical solution does not ensure a discrete adjacency matrix $\tilde{\boldsymbol{A}}$. Thus, we adopt a thresholding strategy to discretize $\tilde{\boldsymbol{A}}$, setting entries below and above the threshold to 0 and 1, respectively. The threshold is linearly searched between the maximum and minimum entries of $\tilde{\boldsymbol{A}}$, aiming to minimize the density difference between the mixup graph and the original graphs. The mixup graphs are added to the training set for augmentation, and the training, validating and testing procedures follow the traditional paradigms.

---

**Algorithm 3** The whole mixup procedure

---

1: **Input:** Training set $\mathcal{G} = \{(G_i, y_i)\}_i$, mixup ratio $\beta$
2: **Output:** Augmented training set $\tilde{\mathcal{G}}$
3: $N = |\mathcal{G}|$, $N_y = |\{y_i\}_i|$, $\tilde{\mathcal{G}} = \mathcal{G}$
4: **for** $i \neq j$ in range $(N_y)$ **do**
5:     **for** $k$ in range $(2\beta N/N_y(N_y - 1))$ **do**
6:         Randomly sample $G_i = (\boldsymbol{\mu}_i, \boldsymbol{X}_i, \boldsymbol{A}_i)$ from class $y_i$ and $G_j = (\boldsymbol{\mu}_j, \boldsymbol{X}_j, \boldsymbol{A}_j)$ from $y_j$
7:         Generate new mixup graph and its label $\tilde{G}, \tilde{y} = \texttt{FGWMixup}_{(*)}(G_i, G_j)$, $\tilde{G} = (\tilde{\boldsymbol{\mu}}, \tilde{\boldsymbol{X}}, \tilde{\boldsymbol{A}})$ .
8:         Search the threshold $\theta$ to discretize $\tilde{\boldsymbol{A}}$: $\tilde{\boldsymbol{A}} \leftarrow \{1 \text{ if } a_{ij} \geq \theta \text{ else } 0\}_{ij}$, which minimizes the density differences between $\tilde{\boldsymbol{A}}$ and $\boldsymbol{A}_1$ , $\boldsymbol{A}_2$.
9:         $\tilde{\mathcal{G}} \leftarrow \tilde{\mathcal{G}} \cup \{\tilde{G}\}$
10:     **end for**
11: **end for**
12: **return** $\tilde{\mathcal{G}}$

---

# D  Experimental Settings

## D.1  Dataset Information

PROTEINS, NCI1, NCI109, IMDB-BINARY (IMDB-B) and IMDB-MULTI (IMDB-M) are the five benchmark graph classification datasets used in our experiments. When preprocessing the datasets, we remove all the isolated nodes (i.e., 0-degree nodes) in order to avoid invalid empty message passing of MPNNs. We present the detailed statistics of the preprocessed datasets in Table 6, including the number of graphs, the average/median node numbers, the average/median edge numbers, the dimension of node features, and the number of classes.

## D.2  Backbones

We adopt two categories of GNN frameworks as the backbones. The first category is the virtual-node-enhanced Message Passing Neural Networks (MPNNs), including vGCN and vGIN. The second category is the Graph Transformer-based networks, including Graphormer and GraphormerGD. The details of these GNN layers are listed as follows:

- **vGCN** [1]: Graph Convolution Network is developed from the spectral GNNs with 1-order approximation of Chebychev polynomial expansion. The update of features can be regarded as message passing procedure. The graph convolution operator of each layer in the vGCN is

defined as: $\mathbf{X}^{(l+1)} = \sigma(\tilde{\mathbf{A}}\mathbf{X}^{(l)}\mathbf{W}^{(l)})$, where $\tilde{\mathbf{A}}$ is the renormalized adjacency matrix, and $\mathbf{W}$ is the learnable parameters. We apply ReLU as the activation function $\sigma(\cdot)$.

- **vGIN** [2]: Graph Isomorphism Network takes the idea from the Weisfeiler-Lehman kernel, and define the feature update of each layer through explicit message passing procedure. The GIN layer takes the form of: $x_v^{(l+1)} = \mathrm{MLP}^{(l+1)}\left(\left(1 + \epsilon^{(l+1)}\right) \cdot x_v^{(l)} + \sum_{u \in \mathcal{N}(v)} x_u^{(l)}\right)$, where $\epsilon$ is a learnable scalar. We apply two-layer MLP with ReLU activation in our implementation.

- **Graphormer** [39]: Graphormer adopts the idea of the Transformer [62] architecture, which designs a multi-head global QKV-attention across all graph nodes, and encode graph topologies as positional encodings (spatial encoding in Graphormers). A Graphormer layer conducts feature updating as follows:

$$\mathbf{A}^h\left(\mathbf{X}^{(l)}\right) = \mathrm{softmax}\left(\mathbf{X}^{(l)}\mathbf{W}_Q^{l,h}(\mathbf{X}^{(l)}\mathbf{W}_K^{l,h})^\top + \phi^{l,h}(\mathbf{D})\right);$$

$$\hat{\mathbf{X}}^{(l)} = \mathbf{X}^{(l)} + \sum_{h=1}^{H} \mathbf{A}^h\left(\mathbf{X}^{(l)}\right)\mathbf{X}^{(l)}\mathbf{W}_V^{l,h}\mathbf{W}_O^{l,h};$$

$$\mathbf{X}^{(l+1)} = \hat{\mathbf{X}}^{(l)} + \mathrm{GELU}\left(\hat{\mathbf{X}}^{(l)}\mathbf{W}_1^l\right)\mathbf{W}_2^l,$$

where $\phi(\cdot)$ is the spatial encoding module implemented as embedding tables, and $\mathbf{D}$ is the shortest path distance of the graph. $\mathbf{A}$ is the attention matrix.

- **GraphormerGD** [16]: GraphormerGD is an enhanced version of Graphormer, where the graph topologies are encoded with a *General Distance* metric defined by the combination of the shortest path distance (SPD) and the resistance distance [63] (RD). It mainly modifies the calculation of the attention matrix $\mathbf{A}$ in Graphormer:

$$\mathbf{A}^h\left(\mathbf{X}^{(l)}\right) = \phi_1^{l,h}(\mathbf{D}_{sp}, \mathbf{D}_r) \odot \mathrm{softmax}\left(\mathbf{X}^{(l)}\mathbf{W}_Q^{l,h}(\mathbf{X}^{(l)}\mathbf{W}_K^{l,h})^\top + \phi_2^{l,h}(\mathbf{D}_{sp}, \mathbf{D}_r)\right)$$

where $\phi(\cdot)$ is an MLP taking SPD embedding and RD gaussian kernel embedding as inputs.

For all the backbones, we apply the virtual node READOUT approach (which is the official READOUT function of Graphormers) instead of traditional global pooling. We apply 6 network layers and take 64 as the hidden dimension for all the backbones on all datasets. GraphormerGD has a specical model design on PROTEINS dataset, which applies 5 layers and 32 as the hidden dimension due to the GPU memory limitation.

## D.3 Experimental Environment

**Hardware Environment**    The experiments in this work are conducted on two machines: one with 8 Nvidia RTX3090 GPUs and Intel Xeon E5-2680 CPUs, one with 2 Nvidia RTX8000 GPUs and Intel Xeon Gold 6230 CPUs.

**Software Environment**    Our experiments are implemented with Python 3.9, PyTorch 1.11.0, Deep Graph Library (DGL) [44] 1.0.2, and Python Optimal Transport (POT) [45] 0.8.2. The implementation of all the backbone layers are based on their DGL implementations. For the FGW solver, `FGWMixup` uses the official FGW distance solver implemented in POT, and `FGWMixup`$_*$ uses the accelerated FGW algorithm implemented on our own.

## D.4 Implementation Details

For our mixup algorithm `FGWMixup`, we follow [31] and apply the Euclidean distance as the metric measuring the distance of the node features and the graph adjacency matrix as the structural distances between nodes. We sample the mixup weight $\lambda$ from the distribution $\mathrm{Beta}(0.2, 0.2)$ and the trade-off coefficient of the structure and signal costs $\alpha$ are tuned in $\{0.05, 0.5, 0.95\}$. We set the size of the generated mixup graphs as the weighted average of the source graphs, which is $\tilde{n} = \lambda n_1 + (1 - \lambda)n_2$. The maximum number of iterations of `FGWMixup`is set to 200 for the outer loop optimizing $X$ and $A$, and 300 for the inner loop optimizing couplings $\pi$. The stopping criteria of our algorithm are the relative update of the optimization objective reaching below a threshold, which is set to 5e-4. For

FGWMixup∗, we select the step size of MD $\gamma$ from {0.1, 1, 10}. The mixup ratio (i.e., the proportion of mixup samples to the original training samples) is set to 0.25.

For the training of GNNs, MPNNs are trained for 400 epochs and Graphormers are trained for 300 epochs, both using AdamW optimizer with a weight decay rate of 5e-4. The batch size of GNNs are chosen from {32, 128}, and the learning rate is chosen from {1e-3, 5e-4, 1e-4}. Dropout is employed with a fixed dropout rate 0.5 to prevent overfitting. All the hyperparameters are fine-tuned by grid search on validation sets. For a fair comparison, we employ the same set of hyperparameter configurations for all data augmentation methods in each backbone architecture. For more details, please check our code published online at `https://github.com/ArthurLeoM/FGWMixup`.

# E  Further Experimental Analyses

## E.1  Qualitative Analyses

We present two mixup examples of FGWMixup in Figure 4 to demonstrate that augmented graphs by FGWMixup can preserve key topologies of the original graphs and have semantically meaningful node features simultaneously. In Example 1, we can observe that the mixup graph adopts an overall trident structure and a substructure (marked green) from $G_1$, and adopts several substructures from $G_2$ (marked red), finally formulating a new graph sample combined properties from both graphs. In Example 2, we can observe that the mixup graph is quite similar to $G_2$, but breaks the connection of two marked (red arrow pointed) edges and formulates two disconnected subgraphs, which is identical to the overall structure of $G_1$. Moreover, in both examples, we can observe that the preserved substructures are not only topologically alike, but also highly consistent in node features. The two examples demonstrate that FGWMixup can both preserve key topologies and generate semantically meaningful node features.

Figure 4: Two examples of FGWMixup. In each example, the subfigures on the left and middle are the original graphs to be mixed up, denoted as $G_1$ and $G_2$, respectively. The subtitle denotes the mixup ratio $\lambda$. The subfigure on the right is the synthetic mixup graph. The node features are one-hot encoded and distinguished with the feature ID and corresponding color.

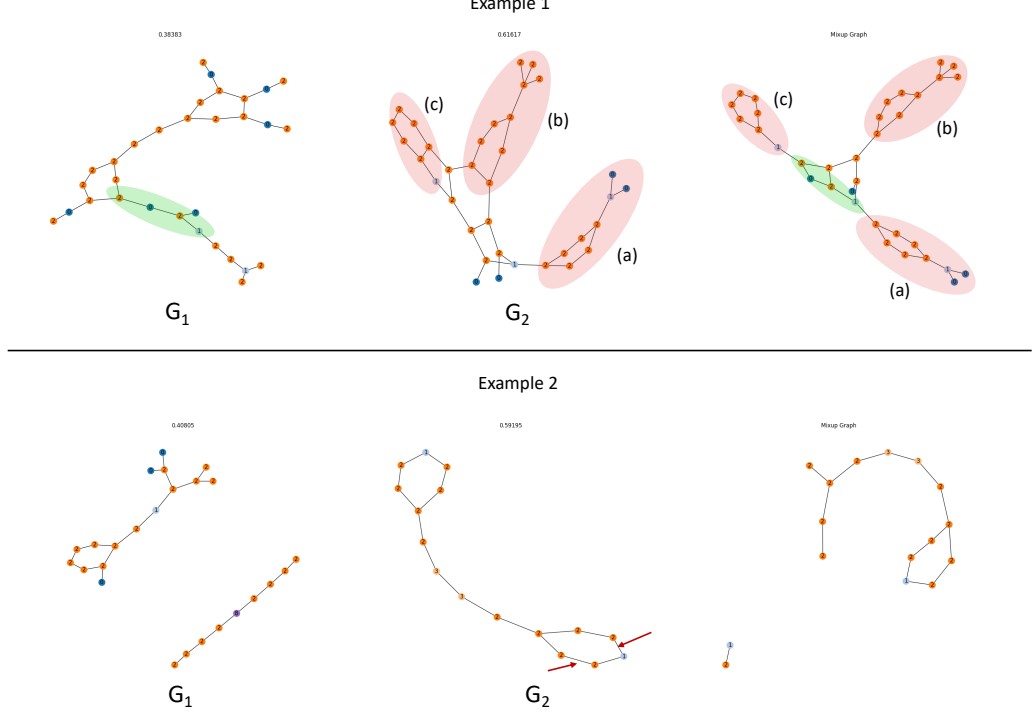

### E.2 Further Discussions on $\mathcal{G}$-Mixup

In this subsection, we conduct further analyses on $\mathcal{G}$-Mixup for a better understanding of its differences from our methods.

**$\mathcal{G}$-Mixup with GW Graphon Estimator**   $\mathcal{G}$-Mixup does not originally apply GW metrics to estimate the graphons (as they introduce in Table 1 and 6 in their paper [20]). In our implementation, we select the USVT method as the graphon estimator for $\mathcal{G}$-Mixup. Yet it is also practical to apply GW metric to estimate graphons as an ablation study. Here, we also provide the experimental results of $\mathcal{G}$-Mixup+GW and $\mathcal{G}$-Mixup+GW* (GW with our single-loop solver) on PROTEINS as shown in Table 7.

| Backbone | $\mathcal{G}$-Mixup | $\mathcal{G}$-Mixup+GW | $\mathcal{G}$-Mixup+GW* | FGWMixup |
|---|---|---|---|---|
| vGIN | 74.84(2.99) | 74.48(2.54) | 74.03(4.46) | **75.02(3.86)** |
| vGCN | 74.57(2.88) | 74.57(3.18) | 74.84(2.15) | **76.01(3.19)** |

Table 7: Experimental results of different graphon estimators of $\mathcal{G}$-Mixup on PROTEINS.

No matter what metrics are applied for the graphon estimation, we want to emphasize that $\mathcal{G}$-Mixup does not model the joint distribution of graph structure and signal spaces, but regards them as two disentangled and independent factors. However, our main contribution and the greatest advantage compared with $\mathcal{G}$-Mixup comes from the joint modeling of graph structure and signal spaces. This also explains why `FGWMixup` can outperform.

**Extend $\mathcal{G}$-Mixup with FGW-based Attributed Graphon Estimator**   In [64], Xu et al. introduce an attributed graphon estimation algorithm, which makes us naturally think of an extended version of $\mathcal{G}$-Mixup. With the attributed graphon, $\mathcal{G}$-Mixup can also consider the joint modeling problem. Though it is not the contribution and the core idea from $\mathcal{G}$-Mixup to address our proposed joint modeling problem, we are also interested in whether this extension can improve the performance of $\mathcal{G}$-Mixup. Hence, we conduct experiments with the extended version of $\mathcal{G}$-Mixup using FGW as the attributed graphon estimator (denoted as $\mathcal{G}$-Mixup+FGW) on PROTEINS and NCI1 datasets, and the results are shown in Table 8.

| Dataset | Backbone | $\mathcal{G}$-Mixup | $\mathcal{G}$-Mixup+FGW | FGWMixup |
|---|---|---|---|---|
| PROTEINS | vGIN | 74.84(2.99) | 74.66(3.51) | **75.02(3.86)** |
| | vGCN | 74.57(2.88) | 74.30(2.85) | **76.01(3.19)** |
| NCI1 | vGIN | 77.79(1.88) | 78.18(1.73) | **78.37(2.40)** |
| | vGCN | 76.42(1.79) | 75.91(1.54) | **78.32(2.65)** |

Table 8: Experimental results of $\mathcal{G}$-Mixup extended with FGW attributed graphon estimator on PROTEINS and NCI1.

We can observe that FGW does not significantly improve the performance of $\mathcal{G}$-Mixup and still cannot outperform our methods. The main reason lies in that the node matching problem remains unsolved using the linear interpolation strategy of two graphons introduced in $\mathcal{G}$-Mixup. Though the intra-class node matching has been done with FGW graphon estimation, the graphons of different classes are not ensured with an aligned node distribution, which requires the inter-class node matching for an appropriate mixup. This is the core reason for the limited performance of $\mathcal{G}$-Mixup series methods.

### E.3 Algorithm Efficiency

In this subsection, we provide additional details regarding the runtime efficiency analyses.

**Further Comparisons between `FGWMixup` and `FGWMixup`$_*$**   Table 9 presents the average time spent on a single iteration of $\pi$ update (i.e, FGW solver convergence) and the average iterations taken for updating $X$ and $A$ until convergence. We can observe that the convergence of a single FGW procedure does not appear to be significantly accelerated. This is because the gradient step

has been taken twice for an alternating projection process, which incurs double time for the gradient calculation (with a complexity of $O(n^3)$, $n$ is the graph size). In contrast, the strict solver only takes one gradient step for each projection. Therefore, despite a faster convergence rate of the relaxed FGW, the extra gradient step makes the overall convergence time of the relaxed FGW solver similar to that of the strict FGW solver. However, a notable improvement is observed in the convergence rate of the outer loop responsible for updating $X$ and $A$, resulting from the larger step of $\pi$ that the relaxed FGW provides, as shown in Prop.1. Considering the two factors above, the overall mixup time has been efficiently decreased by up to $3.46 \times$ with `FGWMixup`$_*$.

| Dataset | Single FGW Iter. Time (s) | | Avg. Outer Loop Converge Iter. (#) | | |
|---|---|---|---|---|---|
| | `FGWMixup`$_*$ | `FGWMixup` | `FGWMixup`$_*$ | `FGWMixup` | Speedup |
| PROTEINS | **0.0181** | 0.0225 | **80.64** | 153.69 | $1.91\times$ |
| NCI1 | 0.0140 | **0.0120** | **54.25** | 170.23 | $3.14\times$ |
| NCI109 | 0.0136 | **0.0123** | **53.12** | 169.98 | $3.20\times$ |
| IMDB-B | 0.0162 | **0.0119** | **20.03** | 95.48 | $4.77\times$ |
| IMDB-M | 0.0081 | **0.0060** | **14.26** | 60.91 | $4.27\times$ |

Table 9: More algorithm execution efficiency details of `FGWMixup` and `FGWMixup`$_*$.

**Comparisons between Our Methods and Compared Baselines** We present the averaged efficiencies of different mixup methods and time spent on training vanilla backbones of each fold on PROTEINS and NCI1 datasets in Table 10. As the efficiency of $\mathcal{G}$-Mixup hugely relies on the graphon estimation approach, we also include $\mathcal{G}$-Mixup with GW graphon estimators (denoted as G-Mixup+GW) in the table.

We can observe that our methods and $\mathcal{G}$-Mixup+GW are slower than the other data augmentation methods. The main reason is that the complexity of calculating (F)GW distances between two graphs is cubic ($\mathcal{O}(mn^2 + nm^2)$)[28], where $m, n$ are the sizes of two graphs. Moreover, when calculating barycenters, we need an outer loop with $T$ iterations and $M$ graphs. In total, the time complexity of mixing up two graphs of size $n$ is $\mathcal{O}(MTn^3)$. `FGWMixup`$_*$ boosts the efficiency by enhancing the convergence rate and reducing the required iterations $T$ (see Table 9 for more details), whereas $\mathcal{G}$-Mixup+GW will have to go over the whole dataset to calculate graphons, which is much more time-consuming than `FGWMixup`.

However, sacrificing complexity to pursue higher performance has been the recent trend of technical development, e.g. GPT-4. Moreover, we believe that the current time complexity of `FGWMixup`$_*$ is still acceptable compared with our performance improvements, as most compared graph augmentation methods cannot effectively enhance the model performance as shown in Table 1.

More importantly, in practice, the main computational bottleneck of (F)GW-based method is the OT network flow CPU solver in the current implementation based on the most widely used POT lib. In other words, GPU-based network flow algorithms have not been applied in current computation frameworks. Moreover, mini-batch parallelization is not yet deployed in POT. However, recent works [65] from NVIDIA have focused on accelerating network flow algorithms on GPU, which may probably be equipped on CUDA and allow a huge acceleration for GW solvers in the near future. Hence, we firmly believe that the fact that our method is not yet optimized in parallel on GPUs is only a temporary problem. Just as some other works (e.g., MLP, LSTM, etc.) that have brought enormous contributions in the past era, they are initially slow when proposed, but have become efficient with fast-following hardware supports.

| Dataset | DropEdge | DropNode | $\mathcal{G}$-Mixup | ifMixup | $\mathcal{G}$-Mixup+GW | `FGWMixup` | `FGWMixup`$_*$ |
|---|---|---|---|---|---|---|---|
| PROTEINS | 0.192 | 0.229 | 6.34 | 2.08 | 2523.78 | 802.24 | 394.57 |
| NCI1 | 0.736 | 0.810 | 10.31 | 5.67 | 9657.48 | 1711.45 | 637.41 |

Table 10: Average mixup efficiency (clock time spent, seconds) of all compared baselines on each fold of PROTEINS and NCI1 datasets.

### E.4 Sensitivity Analysis on Varying Beta Distribution Parameter $k$

We empirically follow the setting in [24] to select the beta distribution parameter $k = 0.2$ where the mixup method is first proposed. We also provide the sensitivity analysis of the Beta distribution parameter $k$ on PROTEINS and NCI1 datasets in Table 11.

| Methods | PROTEINS | | | | NCI1 | | | |
|---|---|---|---|---|---|---|---|---|
| | $k$=0.2 | $k$=0.5 | $k$=1.0 | $k$=2.5 | $k$=0.2 | $k$=0.5 | $k$=1.0 | $k$=2.5 |
| vGIN-FGWMixup | **75.02(3.86)** | **75.02(2.67)** | 74.93(2.93) | 74.39(1.07) | **78.32(2.65)** | 76.37(2.06) | 76.59(2.39) | 77.71(2067) |
| vGCN-FGWMixup | **76.01(3.19)** | 75.47(3.12) | 75.95(2.58) | 74.30(4.12) | **78.37(2.40)** | 78.00(1.40) | 77.98(1.50) | 77.96(1.73) |
| vGIN-FGWMixup$_*$ | 75.20(3.30) | 73.59(2.38) | 73.86(2.81) | **76.10(2.97)** | 77.27(2.71) | 77.25(2.09) | **77.32(1.78)** | 77.01(2.14) |
| vGCN-FGWMixup$_*$ | 75.20(3.03) | 74.29(4.62) | **75.38(3.41)** | 74.21(4.52) | 78.47(1.74) | **78.71(1.49)** | 78.10(1.71) | 77.96(1.02) |

Table 11: Experimental results of different $k$ on PROTEINS and NCI1 datasets.

From the results, we can find that $\mathrm{Beta}(0.2, 0.2)$ is the overall best-performed setting. There are also a few circumstances where the other settings outperform. In our opinion, different datasets and backbones prefer different optimal settings of $k$. However, we should choose the one that is overall the best across various settings.

### E.5 Experiments on OGB Datasets

We also conduct experiments on several OGB benchmark datasets[46], including ogbg-molhiv and ogbg-molbace. We conduct binary classification (molecular property prediction) on these datasets with AUROC (Area Under Receiver Operating Characteristic) as the reported metric. The split of training, validating, and testing datasets are provided by the OGB benchmark. In spite of the existence of edge features in OGB datasets, in this work, both of our GNN backbones and augmentation methods do not encode or consider the edge features. We select vGCN and vGIN (5 layers and 256 hidden dimensions) as the backbones and train the models five times with different random seeds, and we report the average and standard deviation of AUROCs as the results. The dataset information and the predicting performances are listed in Table 12.

From the table, we can observe evident improvements on both datasets and backbones with our methods. FGWMixup$_*$ obtains the best performance on ogbg-molhiv dataset, with 2.69% and 3.46% relative improvements on vGCN and vGIN respectively, and FGWMixup obtains the best on ogbg-molbace dataset with 1.14% and 8.86% relative improvements. Meanwhile, our methods provide a much more stable model performance where we reduce the variance by a significant margin, which demonstrates that our methods can resist the potential noises underlying the data and increase the robustness of GNNs. These additional experimental results further indicate the effectiveness of our methods in improving the performance of GNNs in terms of their generalizability and robustness.

| Stats | | ogbg-molhiv | ogbg-molbace |
|---|---|---|---|
| Graphs | | 41,127 | 1,513 |
| Avg. Nodes | | 25.5 | 34.1 |
| Avg. Edges | | 27.5 | 36.9 |
| Feature Dim. | | 9 | 9 |
| **Backbones** | **Methods** | **ogbg-molhiv** | **ogbg-molbace** |
| vGCN | vanilla | 73.72(2.45) | 76.62(4.59) |
| | DropEdge | 74.07(2.68) | 75.31(4.57) |
| | $\mathcal{G}$-Mixup | 74.65(1.42) | 69.80(6.41) |
| | FGWMixup | 75.24(2.78) | **77.49(2.71)** |
| | FGWMixup$_*$ | **75.70(1.15)** | 76.40(2.45) |
| vGIN | vanilla | 72.61(1.01) | 69.77(1.33) |
| | DropEdge | 72.97(1.61) | 72.68(4.92) |
| | $\mathcal{G}$-Mixup | 74.52(1.58) | 73.74(7.07) |
| | FGWMixup | 72.68(1.47) | **75.95(2.42)** |
| | FGWMixup$_*$ | **75.12(1.02)** | 74.02(2.18) |

Table 12: Statistics and experimental results on OGB benchmark datasets. The reported metrics are AUROCs taking the form of avg.(stddev.).

