# OpenReview forum: "Fused Gromov-Wasserstein Graph Mixup for Graph-level Classifications"
_NeurIPS.cc/2023/Conference — NeurIPS 2023 poster_

### Official Review · Reviewer_Ake3 · 2023-07-04

**Soundness:** 3 good
**Presentation:** 3 good
**Contribution:** 3 good
**Rating:** 5
**Confidence:** 4

**Summary:**

This work presents a novel graph mixup strategy, which focuses on synthesizing a 'midpoint' graph between two graphs based on the graph structure-signal product metric space. The authors utilize the Fused Gromov-Wasserstein (FGW) distance to achieve this and also propose a method to accelerate the computation of FGW distance solvers by relaxing the polytope constraint. The effectiveness of the proposed method is verified through experiments on molecule and social network datasets.

**Strengths:**

- The draft is well-written, and the proposed methods are technically sound  with theoretical analysis.

- Performance improvements compared to the recent baseline are significant.

**Weaknesses:**

- Efficient computation is a key aspect when deploying augmentation strategies in real-world applications. However, Table 3 is not conclusive to me in its current form. Please provide the computation cost for the vanilla model and other baselines to better understand the tradeoff between performance gain and efficiency.

- Lack of qualitative analysis. It remains somewhat unclear whether the synthetic 'midpoint' graphs possess semantic meaning. For instance, in the case of molecules, specific functional groups determine their properties. If these key groups are missing in the synthetic 'midpoint' graphs, it might mislead the network. To provide readers with more insights regarding the 'midpoint' graphs, including real samples of synthetic graphs is recommended.

- The experiments are currently limited to small-scale graph datasets (1K~4K). To demonstrate the applicability of the proposed method to medium/large-scale graphs, conducting experiments on more large-scale datasets such as the OGB benchmark datasets is highly required.

**Questions:**

I noticed that Table 1 shows that the recent state-of-the-art baseline, G-Mixup, performs worse than the vanilla model in many cases. Is there a reason that the authors conjecture for these results?

**Limitations:**

The authors addressed the limitations of their work in the draft.

---

> ### Author Rebuttal · Authors · 2023-08-08
>
> We sincerely appreciate your comments and suggestions. We made every effort to address all the concerns. In the following, we quote your comments and then give our detailed response point-by-point.
>
> > **W1. Please provide the computation cost for the vanilla model and other baselines to better understand the tradeoff between performance gain and efficiency.**
>
> We provide runtime comparisons between our methods and compared baselines as well as the time cost for the vanilla model traning in the public response. Please check **Q1** in the **public response** for details.
>
> > **W2. To provide readers with more insights regarding the 'midpoint' graphs, including real samples of synthetic graphs is recommended.**
>
> Here we provide two mixup examples of FGWMixup in the attachment PDF file in the **public response** for a more comprehensive qualitative analysis. The subfigures on the left and middle are the original graphs to be mixed up, denoted as G1 and G2, respectively. The subtitle denotes the mixup ratio $\lambda$. The subfigure on the right is the synthetic mixup graph. The node features are one-hot encoded and distinguished with the feature ID and corresponding color. In Example 1, we can observe that the mixup graph adopts an overall trident structure and a substructure (marked green) from G1, and adopts several substructures from G2 (marked red), finally formulating a new graph sample combined properties from both graphs. In Example 2, we can observe that the mixup graph is quite similar to G2, but breaks the connection of two marked (red arrow pointed) edges and formulates two disconnected subgraphs, which is identical to the overall structure of G1. Moreover, in both examples, we can observe that the preserved substructures are not only topologically alike, but also highly consistent in node features. These examples demonstrate that **FGWMixup can both preserve key topologies and generate semantically meaningful node features**. We will add the qualitative analysis to the Appendix of our paper.
>
> > **W3. To demonstrate the applicability of the proposed method to medium/large-scale graphs, conducting experiments on more large-scale datasets such as the OGB benchmark datasets is highly required.**
>
> This paper focuses on data augmentation, which mainly tackles the circumstances of data insufficiency. Therefore, it is more persuasive to observe results on small datasets. Furthermore, in former works [17, 18], datasets from TUDataset are widely adopted as the benchmark. Hence, we follow their settings and present the results on those datasets as the main results. However, we also have considered validating the effectiveness of our method on larger datasets, and **we have reported experimental results on large OGB datasets with 40K+ samples** (see Appendix E.2), and our methods also outperform all baselines.
>
> > **Q1. The recent state-of-the-art baseline, G-Mixup, performs worse than the vanilla model in many cases. Is there a reason that the authors conjecture for these results?**
>
> On the one hand, **G-Mixup does not consider the joint modeling of graph structures and node features**. As we discussed in the Intro, ignoring the interaction between the graph structure and signal spaces may degrade the quality of graph data augmentation.
>
> Another reason lies in that **the node matching problem remains unsolved using the linear interpolation strategy** introduced in G-Mixup to mix up two graphons. When calculating graphons, G-Mixup only focuses on aligning node distributions of graphs from the same class, but ignores the alignment across different classes. Therefore, the graphons of different classes are not ensured with an aligned node distribution. Thus, simply conducting the Euclidean addition operation on two unaligned graphons is not appropriate.
>
> The two factors together may lead to a meaningless mixup graphon and probably introduce noises to the dataset, thus leading to performance decay.

---

> > ### Comment · Reviewer_Ake3 · 2023-08-17
> > **Acknowledgement to the authors' rebuttal**
> >
> > I appreciate the authors for their response. After carefully checking the authors' rebuttal and considering the comments from other reviewers, I'm pleased to note that my most concerns have been well addressed. I agree with the authors' claim regarding to the computation cost. As a result, I raise my score from 4 to 5. For the final revision, I kindly request the inclusion of further analysis regarding W2.

---

> > > ### Author Response · Authors · 2023-08-17
> > > **Reply to Reviewer Ake3**
> > >
> > > We would like to express our sincere gratitude for your thoughtful reviews and for considering our rebuttal. We will certainly include the further analyses on time complexity, mixup quality, etc. in our final version.
> > >
> > > Thank you once again for your time, effort, and the positive assessment. Your comments have been invaluable in refining our research.

---

### Official Review · Reviewer_WT1n · 2023-07-04

**Soundness:** 2 fair
**Presentation:** 2 fair
**Contribution:** 2 fair
**Rating:** 6
**Confidence:** 5

**Summary:**

Authors study the problem of graph data augmentation for graph-level classifications. They propose a mix-up strategy based on the computation of Fused Gromov-Wasserstein(FGW) “mid-point” (or barycenter) between a pair of graphs from the training dataset. To solve these optimization problems, they adapt a recent single-loop GW solver [28] to the FGW setting (i.e including node features in the OT problem) and provide a theoretical analysis of the resulting algorithm. Finally, they benchmark their novel mix-up strategy against SOTA mix-up approaches on five real-world datasets using 4 different GNN architectures and show that their method achieves better performances than its competitors.

**Strengths:**

- Authors propose an interesting mix-up strategy capable of incorporating both structure and feature information via the FGW distance.
- They investigate two FGW solvers to estimate the inherent soft (attributed) graph matching problems in a novel task of graph dataset augmentation which is also interesting for the active optimization literature on (F)GW.
- They partially extend the analysis done in [28] for GW to FGW.
- They benchmark their FGWmixup with various SOTA mix-up strategies with 4 different backbones on 5 relatively small real-world datasets (small graphs with around 1k to 4k samples). The supplementary material also contains concluding experiments on the ogbg-molhiv datasets containing +40k samples. Plus study the robustness to label corruption of FGWmixup.
- They study different mix-up strategies w.r.t the size of the FGW barycenters with pairwise (local) and global strategies (e.g proportional to the median size in the dataset)
- Overall I find the paper well-written


**Weaknesses:**

- 1. *[addressed by authors]* **The FGWmixup is badly positioned with respect to the G-mixup** and the current literature on GW (resp. FGW) for the estimation of graphons (resp. attributed graphons):
From my understanding, the G-mixup [18] strategy comes down to estimate 1 graphon per class then perform mix-up by sampling a graph from a linear interpolation between a pair of these graphons. [A] actually showed that a GW barycenter better estimates graphon than all methods considered in G-mixup paper. As discussed in [B], we can consider that an analog result holds for estimating attributed graphons using FGW barycenters. These observations highlight the following weaknesses / missing points in authors’ overall reasoning currently illustrated in the papier:


     - a) G-mixup should be benchmarked using GW solvers which correspond to the FGW solvers studied by the authors. Note that [A] rather studied the Proximal Point solver introduced in [C], which seems to lead to lower performances on several tasks than the single-loop solver in [28] which coincides with your FGWmixup*.

     - b) The extension of the G-mixup strategy using FGW instead of GW should also be discussed and benchmarked. In the same spirit, an ablation study with $\alpha = 1$ should be considered for the FGWmixup. Basically, I would expect from your paper to identity what is better between a global mix-up strategy (G-mixup) or a pairwise one (FGWmixup) for data augmentation. From my point of view, the first one is favorably biased by a class (as a unique barycenter represents a class) but in the meantime the generated graphs are poorly representative of the graph dataset manifold as only one graphon is taken per class. Whereas your FGWmixup is not really biased by a class (potentially too local) but the generated graphs are intrinsically more representative of the input graphs.

- 2. *[partially addressed by authors, the effect of removing marginals' cost is still unclear]* The single loop solver for FGW problems used in Algorithm 2, should be better positioned w.r.t to the one in [28] for GW. It is an exact adaptation from [28] adding the linear OT cost from node features. Moreover you propose to solve for an equivalent problem, where you removed the fixed terms on the marginals. This should be clear in the paper.

     - a) As you remove fixed terms on the marginals, you solve an equivalent problem to get an OT solution. De facto, equation 4 is wrong as you do not consider exactly the gradient of FGW. Make this clearer in the paper please (maybe via different notations for both objective functions).

     - b) In the spirit of [28, Table 7], the feasibility error i.e errors on marginals should be benchmarked across used solvers. I’m worried that removing the marginal terms in the FGW cost makes this feasibility error problem worse. Also this analysis should be completed with an analysis of the estimated FGW distances across used FGW solvers, over a dataset considered in the mix-up experiments.

     - c) The theoretical results, especially proposition 2, are fairly easily proven by following [28]. This reduces the credit we can give to your theoretical contributions.

- 3. [*partially addressed by the authors, the extension to full learning of the 'midpoint' distribution requires more in-depth analysis and will be the subject of future work.*] On the form there are still some points which can be easily improved:

     - a) L93. Mu is not a probability measure, but a probability vector. + L95, references for ‘normalized’ degree distributions ?

     - b) FGW (semi-)metric properties hold on the space of metric spaces quotiented by the notion of strong-isomorphism, like graphs depicted by distance matrices and invariant to permutations. Otherwise, following [D] it can be extended to any graph representations but w.r.t the notion of weak-isomorphism, hence FGW rather defines a pseudo-metric.

     - c) I find the optimization parts confusing and heavy to read. I believe that their readability could be clearly improved by referring efficiently to specific steps in the algorithm and/or specific updates, instead of the confusing nested loop vocabulary.

     - d) I got confused with Equation 2 being a nested bi-level optimization problem, could you further elaborate ? In my opinion it is more simple than that knowing that Atilde and Xtilde are not involved in the coupling constraints.

     - e) Equation 2. In practice it seems that you do not consider learning the masses of the barycenter which removes a degree of freedom in your model. Learning those has actually been showed to be beneficial for many graph representation learning tasks (see [51, 52, 54]). Could you further justify your choice ? and I guess address that as a current limitation of your work. (Considering that it is not difficult to deduce a subgradient for the masses when using bregman projection based solvers.)

     - f)  I believe that the paragraph ‘Effects of Mixup Graph Sizes’ could also take into consideration [E] where graph size also is a current limitation and does not exactly fall into the mentioned problem of graph size generalization. I believe that the latter problem mostly focuses on the question: can we generalize to big graphs when learning on small graphs ? (maybe I’m wrong)

[A] Xu, Hongteng, et al. "Learning graphons via structured gromov-wasserstein barycenters." Proceedings of the AAAI Conference on Artificial Intelligence. Vol. 35. No. 12. 2021.

[B] Xu, Hongteng, et al. "Learning Graphon Autoencoders for Generative Graph Modeling." arXiv preprint arXiv:2105.14244(2021).

[C] Xu, Hongteng, et al. "Gromov-wasserstein learning for graph matching and node embedding." International conference on machine learning. PMLR, 2019.

[D] Chowdhury, Samir, and Facundo Mémoli. "The gromov–wasserstein distance between networks and stable network invariants." Information and Inference: A Journal of the IMA 8.4 (2019): 757-787.

[E] Brogat-Motte, Luc, et al. "Learning to predict graphs with fused Gromov-Wasserstein barycenters." International Conference on Machine Learning. PMLR, 2022.

*Update after rebuttal : The authors have fully or partially addressed most of my concerns through their rebuttals and discussions. Considering that the authors have undertaken to amend the paper and supplementary material accordingly, I increase my grade from 4 (borderline reject) to 6 (weak accept).*


**Questions:**

I refer the authors to the weaknesses' list above for first suggestions and questions. Follows some questions/suggestions to potentially make the paper clearer:

Q1. Could you please explicit the thresholding method used to deduce new sample from the FGWmixup strategy ?

Q2. Could you report an ablation study with respect to the alpha parameter in FGW for the validated values {0.05, 0.5, 0.95} ?


**Limitations:**

I think discussing the weaknesses I've mentioned and answering my questions detailed above would help identify and work around the other limitations of their current work. If the authors manage to address these, I will happily increase my rating.

 This work has no negative societal impact.

---

> ### Author Rebuttal · Authors · 2023-08-09
>
> We sincerely appreciate your comments and suggestions. We made every effort to address all the concerns. In the following, we give our detailed response point-by-point. W denotes Weakness, and Q denotes Questions.
>
> - **W1:**
>
>   - a) We want to point out the fact that G-Mixup does not apply GW metrics to estimate the graphons (as they introduce in Table 1 and Table 6 in their paper [18]). In our implementation, we select the USVT method as the graphon estimator for G-Mixup. Indeed, it is also practical to apply the GW metric for graphon estimation. We also provide the experimental results of G-Mixup+GW and G-Mixup+GW*(single-loop solver) on PROTEINS in Table 1 of the PDF file from the **public response**. No matter what metrics are applied for graphon estimation, we want to emphasize that G-Mixup does not model the joint distribution of graph structure and signal spaces, and regard them as two disentangled and independent factors. However, **our main contribution and the greatest advantage compared with G-Mixup comes from the joint modeling of graph structure and signal spaces**. This also explains why FGWMixup can outperform.
>
>   - b) Just as the reviewer pointed out, G-Mixup can be extended with the attributed graphon using FGW estimator introduced in [B]. With the attributed graphon, G-Mixup can also consider the joint modeling problem. However, this is **NOT** the contribution and the core idea from G-Mixup, and we think **the comparison between our method with the original G-Mixup is already sufficient to prove the effectiveness of our core idea – solving the joint modeling problem with FGW**. Yet, we are also interested in the comparison of the sample-wise and class-wise mixup methods just like the reviewer. Hence, we conduct experiments with the extended version of G-Mixup with FGW graphon estimator, and the results are shown in Table 1 of the PDF file from the **public response**. We can observe that FGW does not significantly improve the performance of G-Mixup and still cannot outperform our methods. The main reasons lie in that the node matching problem remains unsolved using the linear interpolation strategy of two graphons introduced in G-Mixup.  Though the intra-class node matching has been done with FGW graphon estimation, **the graphons of different classes are not ensured with an aligned node distribution**. Simply conducting addition on two unaligned graphons is inappropriate.
>
>   - c) As for the ablation study of $\alpha$=1 for FGWMixup, we will introduce it in our answer to Q2.
>
> - **W2:**
>
>   - a) After removing the constant terms in the FGW equations, we indeed should use another notation for this new optimization objective, and the gradient is calculated on this new objective instead of the original FGW. We will clarify the notation in the paper accordingly.
>
>   - b) For the analysis on infeasibility, please refer to **Q2** in the **public response** for details.
>
> - **W3:**
>
>   - a) Sorry for the mistake, and we will correct $\mu$ as a probability vector. The normalized degree distribution is introduced in [C] (also [49] in our paper), and we will add this reference where we introduce this distribution.
>
>   - b,c) We will clarify the pseudo-metric property of FGW according to [D] in the paper and change the expression of the loop vocabulary to specific steps in the algorithm.
>
>   - d) If we take Eq.(1) into Eq.(2), we can find that Eq.(2) will be a nested minimization problem, where the optimization over $\pi$ involves the calculation of $\tilde{X}$ and $\tilde{A}$ (according to Eq.(1)), and the minimization over $\tilde{X}$ and $\tilde{A}$ relies on the optimal $\pi$. A classic paradigm to solve this problem is to iteratively solve the inner and outer optimization based on the local optima [a]. Actually, we think traditional solvers of (F)GW barycenters [24, 26] are based on this paradigm, and Alg.1 in our work is adapted from those solvers.
>
>   - e) We have considered whether to learn the masses of nodes ($\mu$) in the mixup graph when designing our method. In fact, both choices are practical. For instance, [26] fixes the $\mu$, while [51, 52, 54] adaptively optimize it. However, in our current version, we find that FGWMixup has already outperformed SOTA methods without adaptively learning $\mu$. Considering that this design does not influence the core contribution of our method and the pages are limited, we have not explored the adaptive $\mu$ version in our current paper and regard this as our future work.
>
>   - f) In the paragraph ‘Effects of Mixup Graph Sizes’, we actually do not claim to solve the graph generalization problem, but discuss why the fixed graph size is not a good option in our method. We explain that when we add a large proportion of graphs in a fixed size to the training set, the graph size distribution will be hugely biased and will be bad at generalizing to the test distribution. Especially for small graph sizes (0.5x Median), as we mentioned in the paper, they will make the distribution peaking at small sizes, which will ‘struggle to generalize to larger ones’ and lead to consistently worse performance as shown in Fig.1.
>
> - **Q1:**
>
>   - Our thresholding strategy is introduced in Appendix C. Here we further clarify the concept of ‘density difference’ in this section. The density of a graph G of N nodes is defined by: the number of edges / N(N-1), denoted as den(G). The density difference between the mixup graph $\tilde{A}$ and two original graphs $A_1$, $A_2$ is: |den($\tilde{A}$) – ($\lambda$ den($A_1$) + (1-$\lambda$) den($A_2$))|, and we try to seek a threshold to discretize$\tilde{A}$ for minimizing this density difference.
>
> - **Q2:**
>
>   - Please refer to our response to **Q1** of Reviewer ZCHq for details. (If invisible, we will add them to the comments below)
>
>
> [a] Dempe, Stephan, and F. Mefo Kue, Solving discrete linear bilevel optimization problems using the optimal value reformulation, Journal of Global Optimization 68 (2017): 255-277.

---

> > ### Comment · Reviewer_WT1n · 2023-08-10
> > **Reply to authors**
> >
> > Thank you for your replies, I consider this initial rebuttal compelling and I will increase my grade. Follows some remarks and questions:
> >
> > **About your answers addressed to all reviewers**: The running time comparison is indeed important and should be reported in the supplementary material. Overall if I understood correctly authors' experiments I agree with authors on the fact that the higher computational cost of FGW mixup compared to other augmentation methods is not a huge bottleneck as there is a lot of room to improve those and it constitutes one of the main concern of the current FGW literature (better solvers, better initialisations, GPU friendly solvers etc) while performances' improvements are consistent. Moreover, FGWmixup remains a pre-processing for GNN models that require extensive hyper-parameter tuning and validation. Two small questions to clarify this runtime table:
> >
> >  i) For FGWmixup, does it correspond to the total runtime while performing your FGW mixup on cpus without explicit parallelization over pairs of pairs ? ii) Did you estimate G-mixup + GW graphons using a Block Coordinate Descent ? even in this case, no change needed but I am confident that at this scale(protein or nci1) a better choice would be SGD+Adam both in terms of speed and precision like for GW-based dictionary learning methods.
> >
> > The analysis of feasibility errors is interesting. Could you please complete it by providing the same marginal error measures than in [28] ?
> >
> >
> > **W1** a-b) I agree with authors' rebuttal. Moreover complementary experiments in the PDF in my opinion are important and should be at least reported in the supplementary material and mentioned in the main paper. It shows that FGW in itself (compared to GW) is not enough and proper interpolation schemes are essential. I believe that the natural extension of G-mixup to G-mixup+FGW exhibits well the limitations of linear interpolation between attributed graphons and even more on attributed graphon estimates (modelling at the core of G-mixup), which would require a much more consequent amount of samples to work well when node features come into play.
> >
> > **W3** e) Indeed I believe that this extension deserves to be discussed in the sense that it would really allow the mixup graphs to fully leverage FGW abilities providing non-uniform nodes relative importance which would may be well merged with novel graph pooling techniques e.g weighted means or OT-based pooling.
> >
> > f) I did not want to express that you were claiming to solve the graph generalization problem. In FGW-based structured prediction (cf [E]), the question of predicting graphs with proper sizes is also omnipresent. In both mixup and [E] contexts, one wants to seek a good understanding of the graph data manifold that includes this notion of graph sizes. So indeed, median sizes and so on seem clearly too limited and convex combination of sizes as you suggested is a good idea, but the questions of existence of optimal sizes and whether it coincides with your convex combination remain opened and interesting.
> >
> > Q1. Some thresholding methods were also used e.g in the original paper of FGW to reconstruct shortest path matrices from barycenters, or in the semi-relaxed GW paper for graph completion but perhaps more with a view to reconstruction. To mimic the latter, could you please compare your method benchmarking L2 differences with the following method (detailed only for GW): Denote the GW barycenter $C_{\lambda}$ between $C_1$ and $C_2$ with corresponding OT plans $T_1$ and $T_2$. i) Find best thresholds $\rho_i$ minimising $||C_i - (T_i C_{\lambda} T_i^\top  > \rho_i)||$. ii) compute the discrete $\tilde{C}_{\lambda}$ with threshold $\lambda \rho_1 + (1-\lambda) \rho_2$.
> >
> > Something that bothers me is: if you have to modify the structure of the FGW barycenter so that a given GNN model can read it, why not modifying the barycenter node features too so that both thresholded and row barycenters are as close as possible in the FGW sense ?

---

> > > ### Author Response · Authors · 2023-08-12
> > > **Reply to Reviewer WT1n**
> > >
> > > We genuinely appreciate your kind and timely response, and we address your further concerns point-by-point as follows:
> > > - **Questions about the runtime table:**
> > >   - i): Yes, we report the total runtime of FGWMixup on CPUs without parallelization over graph pairs.
> > >   - ii): Yes, we still use the block coordinate descent method for a fair comparison.
> > >
> > > - **Questions about the feasibility error analysis:**
> > >
> > >     Sure, we also provide the L2 norm of the differences between the row and column marginals given by the two solvers on PROTEINS.
> > > | Row marginal L2 diff | Column marginal L2 diff |
> > > |-------------|-------------|
> > > | 1.478e-9(4.448e-9)      | 0.0016(0.0023)           |
> > >
> > >   From the results, we can also observe that the differences between the marginals are quite small, demonstrating that the single-loop solver will not lead to huge infeasibility.
> > >
> > > - **W1 a-b)** We are happy that the complementary results and our explanation helps. We will certainly add them to our paper to better clarify that a proper interpolation scheme is essential for graph augmentation.
> > >
> > > - **W3 e)** We have attempted to add the extension of adaptive optimization on $\mu$. However, it is not that practical to incorporate this design directly with our current Block Coordinated Descent optimization framework. There are two main reasons: 1) the optimization w.r.t. $\mu$ is constrained with the simplex and does not have an analytic solution as X and A do. Hence, the optimization requires extra Proximal Gradient Descent iterations in the outer loop, making the time complexity significantly higher. 2) In graph dictionary learning [51], $\mu$ is adaptively learned from the whole dataset with sufficient samples, whereas in the mixup scenario, $\mu$ can only be adjusted through two samples. This design may dramatically enlarge the degree of freedom when solving the mixup problem and probably lead to unstable solutions with insufficient samples. Furthermore, [54] does not adaptively learn $\mu$ as they have relaxed the simplex constraints on the target marginal. [52] learns $\mu$ through supervision from downstream tasks, which is not applicable for data preprocessing. In a nutshell, we have not yet discovered a proper method to accomplish the adaptive optimization on $\mu$. Though most previous works [24, 26] assume a known and fixed $\mu$ when solving the barycenter, we still believe it is indeed a valuable question to be explored. We will do more research on this topic in future work.
> > > - **W3 f)** Sorry for our misunderstanding. This indeed is an interesting question. We think the existence of optimal sizes hugely relies on the generation process of the graphs. Only by assuming that graphs are generated from a stationary stochastic process, the optimal sizes can be estimated with proper prior and sufficient data. However, estimating the optimal size for a generic random graph process can be extremely hard, which is not the main focus of this work. On the other hand, our strategy has been validated as effective so far, and we will explore the optimal graph size problem in the future.
> > > - **Q1.**
> > > 1) We guess there might be some typos in the thresholding method that you introduced. The symbol $\rho_i$ is designed to threshold $T_i C_\lambda T_i^{\top}$, which might be inappropriate to discretize $C_\lambda$. Moreover, $C_i$ and $T_i C_\lambda T_i^{\top}$ seem not to be in the same metric space. We also surveyed the two thresholding methods that you mentioned [24, 54]. We use the one in [54] to compare with ours and benchmark on the NCI1 dataset with 500 mixup pairs. The following metrics are reported: 1) averaged L2 of the difference matrix (L2), 2) averaged percentage of non-zero entries in the difference matrix (non-zero%), and 3) percentage of identical matrices given by two methods (identical%).
> > > | L2    | Non-zero\% | Identical\%  |
> > > |-------|-----------|-------------|
> > > | 1.925 | 0.69%   | 40.6%   |
> > >
> > >     We can observe that the two thresholding methods present quite similar results. Hence, we think that our thresholding method is empirically reasonable and effective.
> > >
> > > 2) Like the original FGW paper, the intention of thresholding is to make the structural cost matrix a discrete graph adjacency matrix that can be read by GNNs. However, current node features can already be read by GNNs, thus we only use the threshold discretization to approximate the graph structure without changing the node feature.
> > >
> > >     We appreciate for providing a motivating insight to make the thresholding stricter in the FGW sense. It is true that modifying node features based on current discrete structures can make it closer to the ideal optimum of the FGW barycenter. Yet, this may incorporate extra rounds of optimization on node features with more computation costs. We will consider further optimizing this thresholding method in our future work. But this does not affect our contribution to the core problem we addressed, that is, realizing joint modeling in graph augmentation.

---

> > > > ### Comment · Reviewer_WT1n · 2023-08-15
> > > > **Reply to authors 2**
> > > >
> > > > Thank you for your detailed answers.
> > > >
> > > > Q1. To informally provide the intuition: considering that GW via $T_i$ aims at quantifying the minimal distortion to go from $C_i$ to $C_{\lambda}$, we could use $T_i$ as a projection map to obtain the footprint of $C_{\lambda}$ onto $C_i$ via $T_i C_{\lambda} T_i^\top$. That is why I was suggesting to find the threshold $\rho_i$ that minimizes $|| C_i - T_i C_{\lambda} T_i^\top ||$ (or the product being adequately rescaled using the barycenter masses even if it should not really change the finale outcome with uniform distributions). And then threshold the barycenter $C_{\lambda}$ using the knowledge of $\rho_1$ and $\rho_2$, leveraging the closed-form formula on $C_{\lambda}$. But indeed, this suggestion like others remain heuristics which are not yet that well understood and the discrete optimisation problem inherent to these thresholding strategies falls out of the scope of this paper. Thank you for the sanity check.

---

> > > > > ### Author Response · Authors · 2023-08-15
> > > > > **Reply to Reviewer WT1n #2**
> > > > >
> > > > > Thank you for the detailed clarification. In our understanding of your suggestion, $\rho$ can be the threshold discretizing $C_\lambda$ that minimizes $|| C_i \* \mu \mu^\top - T_i C_\lambda T_i^\top ||$. We think this idea is practical. But indeed, as you mentioned, this idea is also kind of heuristic, and we regard the exploration of this question as an interesting future work.
> > > > >
> > > > > In all, we genuinely appreciate your efforts exerted on our work and all the detailed and constructive advice. We are wondering if we have settled all your doubts and concerns with our responses. We are looking forward to your further reply.

---

> > > > > > ### Comment · Reviewer_WT1n · 2023-08-15
> > > > > > **Reply to authors 3**
> > > > > >
> > > > > > Thank you for your answers.
> > > > > >
> > > > > > Overall, I find your rebuttal convincing. From an empirical point of view, the paper is solid, but I consider that there is considerable room for improvement from a theoretical point of view to pave the way for new mix-up schemes based on OT. Therefore, I increase my rating from 4 (borderline reject) to 6 (weak accept).

---

> > > > > > > ### Author Response · Authors · 2023-08-16
> > > > > > > **Reply to Reviewer WT1n #3**
> > > > > > >
> > > > > > > I would like to express my sincere gratitude for your thoughtful and profound review and for considering our rebuttal. I greatly appreciate your recognition of the improvements and contributions made to our manuscript. Your valuable feedback has undoubtedly enriched the quality of our work.
> > > > > > >
> > > > > > > Thank you once again for your time, effort, and the positive assessment. Your guidance and insights have been invaluable in refining our research. We remain committed to addressing any remaining concerns and ensuring a better quality of our work.

---

### Official Review · Reviewer_ZCHq · 2023-07-04

**Soundness:** 2 fair
**Presentation:** 3 good
**Contribution:** 2 fair
**Rating:** 4
**Confidence:** 3

**Summary:**

This paper proposes a new graph data augmentation method for graph-level classifications. To address the limitation of existing methods, the authors consider the joint interaction between the graph structure and node features by finding an optimal inter-graph node matching strategy. Furthermore, the authors introduce a relaxed FGW solver to accelerate the proposed method FGWMixup. Experiments show that the FGWMixup outperforms multiple baselines on five datasets using different GNN backbones.

**Strengths:**

Overall the paper is well-written and easy to read.

Using the Fused Gromov-Wasserstein distance metric space seems to be an interesting solution for mixing graphs.

The authors provide theoretical analysis for accelerating FGWMixup by improving the convergence rate.

**Weaknesses:**

My major concern is about the claim that most existing graph data augmentation methods only consider one of the graph structure space and node feature space. There exist many graph augmentation methods (adversarial augmentation, local graph augmentation, automated graph augmentation and etc). Many of them change both graph structure and node features to generate augmented graphs. I don't think this work is the only one considering the joint modeling problem.

Besides, the authors claim the importance of the interaction between graph structure space and node features space. I think more explanations/analyses are needed to show that augmented graphs by FGWMixup can preserve key topologies of the original graphs and have semantically meaningful node features simultaneously.

For the experiments, the improvement of the performance is not significant compared to the high std. I would suggest the authors include experiments on the OGB benchmark(https://ogb.stanford.edu/docs/graphprop/) since the std on OGB is typically small compared to TUDataset. Furthermore, except for some graph mixup methods, the authors only compare with DropEdge and DropNode. It would make the experimental results more convincing if the authors can include some advanced graph data augmentation methods.



**Questions:**

There are multiple hyperparameters in the proposed FGWMixup, ex. $\alpha$ in equation 1. How to choose the hyperparameters? How sensitive is the performance gain to the hyperparameter tuning?

The authors provide experiments about the running time between FGWMixup and FGWMixup*. How does FGWMixup compare to the other baselines? Does the improvement of the performance come from a much higher computational cost?



**Limitations:**

Yes. The authors have discussed the limitation of their method.

---

> ### Author Rebuttal · Authors · 2023-08-08
>
> We sincerely appreciate your comments and suggestions. We made every effort to address all the concerns. In the following, we quote your comments and then give our detailed response point-by-point.
>
> > **W1. Many existing works change both graph structure and node features to generate augmented graphs. I don't think this work is the only one considering the joint modeling problem.**
>
> We want to emphasize that **we do NOT claim that existing methods only consider one of the graph structure and the node feature**, but express that **most existing works regard them as two disentangled perspectives and consider the augmentation on the two parts independently (separately)**. This means most works do not consider the effects of node features when generating new graph structures, and vice versa. We provide two examples (G-Mixup, ifMixup) in the Intro, and more are listed in the Related Works. However, as we mentioned in L.45-46 in the Intro, the graph structures and node features are correlated with each other, and it is essential to depict this correlation while generating new graph samples. Hence, we propose to consider the joint modeling problem to enhance the quality of graph augmentation.
>
> > **W2. More explanations/analyses are needed to show that augmented graphs by FGWMixup can preserve key topologies of the original graphs and have semantically meaningful node features simultaneously.**
>
> We have presented two mixup examples in the attachment PDF file in the **public response**. In Example 1, we can observe that the mixup graph adopts an overall trident structure and a substructure (marked green) from G1, and adopts several substructures from G2 (marked red), finally formulating a new graph sample combined properties from both graphs. In Example 2, we can observe that the mixup graph is quite similar to G2, but breaks the connection of two marked (red arrow pointed) edges and formulates two disconnected subgraphs, which is identical to the overall structure of G1. Moreover, in both examples, we can observe that the preserved substructures are not only topologically alike, but also highly consistent in node features. These examples demonstrate that **FGWMixup can both preserve key topologies and generate semantically meaningful node features**. We will add this analysis to the Appendix of our paper.
>
> > **W3. It would be more convincing to include larger datasets and more data augmentation baselines.**
>
> This paper focuses on data augmentation, which mainly tackles the circumstances of data insufficiency. Therefore, it is more persuasive to observe results on small datasets. Furthermore, in former works [17, 18], datasets from TUDataset are widely adopted as the benchmark. Hence, we follow their settings and present the results on those datasets as the main results. However, we also have considered validating the effectiveness of our method on larger datasets, and **we have reported experimental results on large OGB datasets with 40K+ samples** (see Appendix E.2) and our methods also outperform all baselines.
>
> For the compared baselines, we think mixup-based methods are the state-of-the-art of graph augmentation. Former works such as ifMixup and G-Mixup have included some other graph augmentation methods as baselines (e.g., Subgraph, NodeAttrMasking), and have proven their superiority to those methods. Therefore, we think it is sufficiently convincing and persuasive to adopt SOTA mixup methods as the compared baselines.
>
> > **Q1. How to choose the hyperparameters? How sensitive is the performance gain to the hyperparameter tuning?**
>
> As we introduced in Appendix D.4. L674, the hyperparameters are selected by grid search on validation sets. We have provided the analysis on hyperparameters such as graph sizes and GNN depths in Section 3.3. Here we additionally provide the sensitivity analysis w.r.t. $\alpha$ valued from {0.05, 0.5, 0.95, 1.0}. Note that $\alpha =1.0$ falls back to the case of GW metric where node features are not incorporated.
>
> Results on PROTEINS:
> | $\alpha$ | GIN (FGWMixup) | GCN (FGWMixup) | GIN (FGWMixup*) | GCN (FGWMixup*) |
> |----------|----------------|----------------|-----------------|-----------------|
> | 0.95     | 0.7502(0.0386) | **0.7601(0.0319)** | **0.7520(0.0330)**  | **0.7520(0.0303)** |
> | 0.5      | **0.7538(0.0258)** | 0.7547(0.0356) | 0.7457(0.0330)  | 0.7457(0.0352)  |
> | 0.05     | 0.7486(0.0240) | 0.7493(0.0274) | 0.7439(0.0301)  | 0.7484(0.0316)  |
> | 1.0      | 0.7457(0.0262) | 0.7440(0.0357) | 0.7394(0.0386)  | 0.7466(0.0291)  |
>
> Results on NCI1:
> | $\alpha$ | GIN (FGWMixup) | GCN (FGWMixup) | GIN (FGWMixup*) | GCN (FGWMixup*) |
> |----------|----------------|----------------|-----------------|-----------------|
> | 0.95     | **0.7832(0.0265)** | **0.7837(0.0240)** | **0.7727(0.0271)**  | 0.7847(0.0174)  |
> | 0.5      | 0.7742(0.0193) | 0.7793(0.0168) | 0.7723(0.0247)  | 0.7766(0.0148)  |
> | 0.05     | 0.7762(0.0237) | 0.7800(0.0100) | 0.7659(0.0214)  | **0.7893(0.0191)**  |
> | 1.0      | 0.7591(0.0293) | 0.7727(0.0092) | 0.7720(0.0169)  | 0.7771(0.0197)  |
>
> From the results, we can observe that 1) when FGW falls back to GW ($\alpha$ =1), **where node features are no longer taken into account, the performance will significantly decay** (generally the worst among all investigated alpha values). This demonstrates the importance of solving the joint modeling problem in graph mixup tasks. 2) $\alpha$=0.95 is the best setting in most cases. This empirically implies that **it is better to conduct more structural alignment in graph mixup**. In practice, we set $\alpha$ to 0.95 for all of our reported results. We will add this analysis to the paper.
>
> > **Q2. How does the runtime of FGWMixup compare to the other baselines? Does the improvement of the performance come from a much higher computational cost?**
>
> We provide runtime comparisons between our methods and baselines in the public response. Please check **Q1** in the **public response** for details.

---

> ### Comment · Area_Chair_vRuB · 2023-08-19
>
> Dear Reviewer,
>
> The authors have provided a comprehensive response to your review. Could you please confirm if it addresses your concerns? With the rebuttal deadline fast approaching, we would greatly appreciate your feedback before then.
>
> Thank you for your time and consideration.

---

> > ### Comment · Reviewer_ZCHq · 2023-08-19
> > **Reply to authors**
> >
> > Thanks for the rebuttals.
> >
> > I have some remaining questions.
> >
> > For the results on ogbg-molhiv dataset, why the performance of the baseline models vGCN and vGIN are much lower than the OGB leaderboard? Based on my previous experiments, ignoring edge features should not cause such a significant performance drop. Besides, in Table 6 of the Appendix, I didn't see a significant reduction in the variance when using vGIN as backbones, I believe the authors made a false claim in lines 706-710.
> >
> > For the baselines, I agree that previous mixup methods compare with other simple data augmentation methods such as Subgraph, and NodeAttrMasking. However, there is no comparison between mixup methods and other advanced methods such as [1][2].
> >
> >
> > [1] Kong, Kezhi, et al. "Flag: Adversarial data augmentation for graph neural networks." CVPR 2022.
> >
> > [2] You, Yuning, et al. "Graph contrastive learning automated." ICML, 2021.
> >
> > [3] Luo, Youzhi, et al. "Automated data augmentations for graph classification." ICLR 2023.

---

> > > ### Author Response · Authors · 2023-08-20
> > > **Reply to Reviewer ZCHq**
> > >
> > > Thank you for your reply, and we address your further concerns point-by-point as follows:
> > >
> > > > **Q1. The difference in the vGIN/vGCN performances between ours and OGB leaderboard.**
> > >
> > > Except for ignoring edge features, the difference in the experimental results can be attributed to the experimental environments. In our experiments, vGIN and vGCN are implemented with dgl library, which is not comparable to the results in the OGB leaderboard using the torch_geometric lib. More importantly, the vanilla model is consistently deployed for every compared graph augmentation method. **We suggest that it is the performance gain of each augmentation method that should be paid more attention to, instead of the absolute performance of the vanilla model.**
> > >
> > > > **Q2. The std reduction of vGIN is not significant.**
> > >
> > > Indeed, the std of vanilla vGIN is quite small, but the performance of vanilla vGIN is also relatively low. Thus, we believe that the vanilla vGIN is stably influenced by the underlying noises. In contrast, **our method can provide a 2.5%+ performance gain with only a 0.01% std increase. This shows that our method can help resist those potential noises and increase the model robustness.** Moreover, we find that although the investigated augmentation methods can improve the performance of vGIN, they also lead to much higher variance. However, our method can **effectively improve the performance and simultaneously keep the variance at a low level**. From this perspective, we can also reach a conclusion that our method is better at guaranteeing model robustness.
> > >
> > > > **Q3. Some other baseline methods?**
> > >
> > > In response to your query regarding the baseline comparison, we want to assure you that our choice of baseline models is consistent with standard evaluation protocols of the previous graph mixup works (such as G-Mixup, ifMixup). Our baselines are selected from those used in the previous works and, of course, the previous mixup works themselves. Hence, this approach has already provided a meaningful assessment of the effectiveness of our method within the existing context.
> > >
> > > We acknowledge the articles you mentioned, and we have taken into account their comparisons. However, not all the mentioned articles should be considered. For example, [2] is a graph contrastive learning framework whose negative samples comes from some augmentation methods (including DropNode, Subgraph, etc that we have considered or mentioned). We believe our method can be incorporated into their work, but should NOT be compared with theirs.
> > >
> > > We also want to claim that **the core contribution of this work is a novel graph mixup method that jointly models the interaction between graph signal space and structure space**. We think **it is more necessary and convincing to compare our methods with the SOTA mixup methods to validate our contribution**.
> > >
> > > In conclusion, we believe that our approach provides a well-rounded evaluation framework that captures the essence of the problem while maintaining consistency with previous evaluation practices. We hope this explanation clarifies our contribution and its alignment with existing evaluation standards.

---

### Official Review · Reviewer_JSkL · 2023-07-05

**Soundness:** 3 good
**Presentation:** 2 fair
**Contribution:** 2 fair
**Rating:** 6
**Confidence:** 3

**Summary:**

This paper addresses a gap in graph data augmentation for graph-level classifications, where existing methods mainly focus on augmenting graph signal space and graph structure space separately, overlooking their mutual interactions. The authors formulate the issue as an optimal transport problem that considers both graph structures and signals, and propose a novel graph mixup algorithm called FGWMixup. FGWMixup seeks the "midpoint" of source graphs in the Fused Gromov-Wasserstein (FGW) metric space. The authors further introduce a relaxed FGW solver to improve the scalability and performance of FGWMixup, and experimental results across five datasets and various GNN backbones demonstrate its effectiveness in enhancing the generalizability and robustness of GNNs.

**Strengths:**

S1. Innovation in Graph Mixup Method: The paper introduces FGWMixup, a novel graph mixup method that is formulated as an optimal transport problem. This innovative approach aims to find the optimal graph at the "midpoint" of two source graphs in terms of both graph signals and structures. This unique formulation could potentially provide a more effective way to combine graphs compared to traditional methods.
S2. Comprehensive Evaluation: The paper comprehensively evaluates the proposed method on five widely-used graph classification tasks from the graph benchmark dataset. The use of four different types of backbones for the evaluation further demonstrates the versatility and robustness of the FGWMixup method. This extensive evaluation provides strong evidence of the effectiveness of the proposed method across various tasks and settings.
S3. Theoretical Analysis and Optimization: The paper provides a thorough theoretical analysis of the proposed FGWMixup method, including the convergence of the algorithm and the correctness of FGWMixup. This rigorous theoretical foundation strengthens the credibility of the proposed method. Additionally, using an algorithm to optimize the computation complexity is a commendable effort to address potential efficiency concerns.

**Weaknesses:**

W1. High Time Complexity: The proposed FGWMixup and Accelerated FGWMixup methods have a relatively high time complexity. This could limit their applicability in scenarios where computational resources or time are constrained. The paper would benefit from a more detailed analysis of the time complexity of these methods, including how it scales with the size and complexity of the input graphs.
W2. Lack of Results for Varying Beta Distribution Parameter: The paper needs results showing the impact of changing the beta distribution parameter 'k' on the performance of the proposed methods. This parameter could significantly influence the distribution, and it would be informative to understand how its variation affects the results. This could also provide insights into how to choose the best 'k' for different scenarios.
W3. Effects of Relaxed Projection: The paper could provide more discussion about the potential negative effects due to the relaxed projection. While this approach may improve the algorithm's efficiency, it could also introduce inaccuracies or instability in the results. A deeper exploration of these trade-offs would be beneficial.

**Questions:**

Q1. The algorithm may sacrifice some feasibility due to the relaxed projection. Can you provide some potential effects?
Q2. How does the computational complexity of FGWMixup compare to other graph mixup methods?

**Limitations:**

The paper does not explicitly address the potential limitation of the proposed approach. The authors can test their method on node classification or link prediction tasks.

---

> ### Author Rebuttal · Authors · 2023-08-08
>
> We sincerely appreciate your comments and suggestions. We made every effort to address all the concerns. In the following, we quote your comments and then give our detailed response point-by-point.
>
> > **W1/Q2. Analysis and comparisons of computational complexity:**
>
> We provide runtime comparisons between our methods and compared baselines as well as the computation complexity analysis in the public response. Please check **Q1** in the **public response** for details.
>
> > **W2. Sensitivity analysis on varying beta distribution parameter $k$:**
>
> Empirically, we follow the $k$ setting in [a] where the mixup method is first proposed. We also provide the sensitivity analysis of the Beta distribution parameter $k$ on PROTEINS and NCI1 datasets as follows:
>
> Results on PROTEINS:
> |  k   | GIN (FGWMixup) | GCN (FGWMixup)| GIN (FGWMixup*)| GCN (FGWMixup*)|
> |-----|----------------|----------------|----------------|-----------------|
> | 0.2 | **0.7502(0.0386)** | **0.7601(0.0319)** | 0.7520(0.0330) | 0.7520(0.0303)  |
> | 0.5 | **0.7502(0.0267)** | 0.7547(0.0312) | 0.7359(0.0238) | 0.7429(0.0462)  |
> | 1.0 | 0.7493(0.0293) | 0.7565(0.0258) | 0.7386(0.0281) | **0.7538(0.0341)** |
> | 2.5 | 0.7439(0.0107) | 0.7430(0.0412) | **0.7610(0.0297)** | 0.7421(0.0452)  |
>
> Results on NCI1:
> |  k   | GIN (FGWMixup)| GCN (FGWMixup)| GIN (FGWMixup*)| GCN (FGWMixup*)|
> |-----|----------------|----------------|----------------|-----------------|
> | 0.2 | **0.7832(0.0265)** | **0.7837(0.0240)** | 0.7727(0.0271) | 0.7847(0.0174)  |
> | 0.5 | 0.7637(0.0206) | 0.7800(0.0140) | 0.7725(0.0209) | **0.7871(0.0149)**  |
> | 1.0 | 0.7659(0.0239) | 0.7798(0.0150) | **0.7732(0.0178)** | 0.7810(0.0171)  |
> | 2.5 | 0.7771(0.0267) | 0.7796(0.0173) | 0.7701(0.0214) | 0.7796(0.0102)  |
>
> From the results, we can find that **Beta(0.2, 0.2) is the overall best-performed setting**. There are also a few circumstances where the other settings outperform. In our opinion, different datasets and backbones prefer different optimal settings of $k$, but we should choose the one that is overall the best across various settings.
>
> > **W3/Q1. More exploration of the infeasibility introduced by the relaxed FGW solver:**
>
> First of all, we conduct an experiment to analyze how much infeasibility or inaccuracy has been introduced by our single-loop FGW solver compared with the strict CG solver. We randomly select 1,000 pairs of graphs from PROTEINS dataset and apply the two solvers to calculate the FGW distance between each pair of graphs. The distances of the $i$-th pair of graphs calculated by the strict solver and the relaxed solver are denoted as $d_i$ and $d^{\*}_i$, respectively. We report the following metrics for comparison: i) MAE (mean absolute error of FGW distance, $\frac{1}{N}\sum |d_i - d^{\*}_i|$), ii) MAPE (mean absolute percentage error of FGW distance,  $\frac{1}{N}\sum \frac{|d_i - d^{\*}_i|}{d_i}$), iii) mean FGW distance given by the single-loop solver ($\frac{1}{N}\sum d^{\*}_i$), iv) mean FGW distance given by the strict CG solver ($\frac{1}{N}\sum d_i$), v) L2-norm of the difference between two transportation plan matrices (divided by the size of the matrix for normalization). The results are shown as follows:
>
> | MAE            | MAPE           | mean_FGW | mean_FGW\* | T_diff          |
> |----------------|----------------|----------|-----------|-----------------|
> | 0.0126(0.0170) | 0.0748(0.1022) | 0.2198   | 0.2143    | 0.0006(0.0010)  |
>
> We can observe that the MAPE is only 0.0748, which means the FGW distance estimated by the single-loop relaxed solver is only 7.48% different from the strict CG solver. Moreover, the absolute error is around 0.01, which is quite small compared with the absolute value of FGW distances (~0.21). We can also find that the L2-norm of the difference between two transportation plan matrices is only 0.0006, which means two solvers give quite similar transportation plans. All the results imply that **the single-loop solver will not produce huge infeasibility or make the estimation of FGW distance inaccurate**.
>
> From another perspective, we do not think the infeasibility brings totally negative effects. As we have analyzed in our experiments (Table 1), FGWMixup* even sometimes performs better than FGWMixup. We explain this with the subtle infeasibilities introduced by the relaxed single-loop solver. By incorporating those infeasibilities, FGWMixup* may be able to **occasionally generate more diverse examples, which potentially enlarge the input space, thus bringing opportunities to improve the generalizability of GNN models.**
>
>
> [a] Zhang Hongyi et al., mixup: Beyond Empirical Risk Minimization, ICLR 2018

---

> > ### Comment · Reviewer_JSkL · 2023-08-15
> > **Ack of rebuttal**
> >
> > I thank the authors for the rebuttal. It addressed most of my concerns and I hope these discussions and remedies can be properly incorporated into the final version. I have raised my overall rating.

---

> > > ### Author Response · Authors · 2023-08-16
> > > **Reply to Reviewer JSkL**
> > >
> > > Thanks again for all the constructive suggestions for improving the quality of our work. We will incorporate those discussions and remedies into our final version.

---

### Official Review · Reviewer_Cjiv · 2023-07-13

**Soundness:** 3 good
**Presentation:** 2 fair
**Contribution:** 2 fair
**Rating:** 6
**Confidence:** 4

**Summary:**

In this paper, the authors study a new method "FGWMixup" for graph data augmentation.

For the two input graphs $G_1, G_2$, they propose to construct a synthetic graph $\tilde G$ through optimizing the weighted distance sum (2). To further improve the efficiency of the algorithm, they relax the polytope constraint (row sum and column sum meet the source and target distribution) on the transport matrix $\pi$, to two alternatively enforced constraints (separately on row sum and column sum); they summarize the new solver in Algorithm 2.

They also study the performance of the proposed method/solver through empirical experiments.

**Strengths:**

- originality:
    - While the rough idea to use OT for data augmentation is regular, this paper proposes a different solution and a new relaxed FGW solver, which is novel to me.

- quality:
    - The new method/solver they propose does work and can provide comparable performance to existing mixup methods.

- significance:
    - This paper provides a OT-based mixup method, which can be a good reference for self-supervised learning on graph.

**Weaknesses:**

- quality
    - Some empirical results may be unconvincing due to the small size of datasets. I leave a related comment below.

- clarity:
    - Some concepts are not clearly illustrated. See the questions below. With the unclear illustration, the theoretical justifications for some claims are hard to follow.

- significance:
    - Even with the new approximate solver, I doubt the new FGWmixup is much slower than previous mixup methods, which can make the work less attractive to practitioners.

**Questions:**

1. In solving (2), they "fix the node probability distribution of $\tilde G$ with a uniform distribution" (Line 129). It makes sense, while it can be better to add more justification, either empirically or theoretically.
2. The concept of “Bregman projection” in Line 164, 171 is not well explained.
3. The statement of Proposition 1 is confusing.
        - "Let $\pi_t$ be the Sinkhorn iterations" is unclear. What's the relationship between $\pi_t$ and the following $\pi_{2t}, \pi_{2t+1}$?
        - "Let $\pi^*$ be the unique optimal solution" needs more explanation, optimal w.r.t. what?
        - What's the definition of $\mu_t, \nu_t$?
        - Does the proposition mean the new solver may not converge to $\pi^*$?
4. The statement of Proposition 2 is also confusing.
        - The concept of “FGW function” is not defined.
        - The concept of “normal cone” is not well explained.
        - Actually the follow-up remark after Prop 2 is not very informative. Please provide more details and explain why the distance bound implies FGWMixup∗ "will converge close to the ideal optimum". What's the scale of $\tau/\rho$?
5. The experiments only involve small datasets. Can the authors add some medium to large datasets, and also report the runtime of different mixup methods?

**Limitations:**

N/A.

---

> ### Author Rebuttal · Authors · 2023-08-08
>
> We sincerely appreciate your comments and suggestions. We made every effort to address all the concerns. In the following, we quote your comments and then give our detailed response point-by-point. The references given in numbers correspond to the reference id in our paper, and those given in letters are provided at the end of our response.
>
> > **Q1. Justification on the uniform distribution selection of $\tilde{G}$:**
>
> In most cases where we have no prior knowledge of the node importance, works [a, 24] empirically choose a uniform distribution of $\mu$. There are also some works [51, 52, 54] adaptively optimize it, which incorporate another degree of freedom to the model. Both approaches are reasonable and practical. In our current version, we choose the former one, and we find that FGWMixup has already outperformed SOTA methods with a fixed uniform distribution of $\mu$. Considering that this design does not influence the core contribution of our method and the pages are limited, we have not explored the adaptive $\mu$ version of FGWMixup in our current paper. We regard this as our future work. We will add more explanations and references of this design to the paper for better justification.
>
> > **Q2: The concept of "Bregman Projection":**
>
> We have introduced the Bregman projection $\phi$ is in Appendix A.2 Proposition 3. Specifically, Bregman projection is a function $\phi()$ used to calculate the Bregman divergence $D_\phi(x,y)$. More concretely, in the Mirror Descent algorithm, due to the geometry constraint of the feasible set $\mathcal{X}$, we aim to map $x$ to the dual space and take a gradient step there, thence mapping it back to the primal space (illustrated in Fig.3 in Appendix A.2). The dual space is precisely produced by this Bregman projection, which maps the primal space with $\nabla \phi()$.
>
> > **Q3: Confusions about Proposition 1 - Definition of notations and meaning of the proposition:**
>
> Sorry for the confusion. We clarify the questions as follows:
> 1) *What's the relationship between $\pi_t$ and $\pi_{2t}, \pi_{2t+1}$?* $\\{ \pi_t \\}$ is the sequence of $\pi$ generated from each update step $t$ of Sinkhorn iterations. At the even number steps ($2t$), we optimize the first marginal $\mu$, and at the odd number steps ($2t+1)$, we optimize the second marginal $\nu$. This is precisely the alternately updating procedure in Alg.2.
> 2) *What is $\\pi^{\*}$ optimal w.r.t?* $\pi^*$ is the optimal solution of the FGW distance, which can also be regarded as a function of $\pi$ (i.e., $f(\pi)$), as shown in Appendix A.1.
> 3) *What is the definition of $(\mu_t, \nu_t)$?* As we mentioned in Prop.1, $\pi \in \Pi(\mu, \nu)$. Analogously, $(\mu_t, \nu_t)$ is the marginals of $\pi_t$.
> 4) *The meaning of the proposition?* This proposition demonstrates a faster convergence rate of the single-loop solver. Specifically, the divergences between the optimized marginals $\mu_t, \nu_t$ and the objective marginals $\mu, \nu$ descend quadratically as iteration $t$ goes  ($O(t^{-2})$), which is optimized by the single loop solver. Yet the divergence of strict solver over the joint distribution $\pi$ goes linearly ($O(t^{-1})$).
>
> > **Q4: Confusions about Proposition 2 - Definition of concepts, meaning of the proposition, and the scale of the upper bound:**
>
> Sorry for the confusion. The explanations are as follows:
> 1) *What does the FGW function refer to?*  The FGW function is introduced in Eq.(1), and it can be regarded as a function of $\pi$ ($f(\pi)$). We have introduced this in Appendix A.1.
> 2) *The definition of "normal cone"?* The concept of the normal cone is introduced in Definition 1 in Appendix A.2.
> 3) *What is the scale of $\tau/\rho$ and why will our algorithm "converge close to the ideal optimum"?* Prop.2 gives an upper bound ($\tau/\rho$) of the distance between the optima given by our relaxed solver and the ideal optimum. In fact, as we introduced in L.557 and L.177, the entropic regularization coefficient $\rho$ is actually equivalent to $1/\gamma$, where $\gamma$ is the step size of MD update. $\tau$ is a fixed constant given by various factors (see Appendix B.2). Noted that the step size of MD can be arbitrarily adjusted, **when we select a step size as small as possible, the scale of the upper bound $\tau/\rho$ will be sufficiently small**. This proves that FGWMixup* can converge to the ideal optimum.
>
> > **Q5: Requirements of larger datasets and runtime of different mixup methods:**
>
> 1) This paper focuses on data augmentation, which mainly tackles circumstances of data insufficiency. Therefore, it is more persuasive to observe results on small datasets. Furthermore, in former works [17, 18], datasets from TUDataset are widely adopted as the benchmark. Hence, we follow their settings and present the results on those datasets as the main results. However, we also have considered validating the effectiveness of our method on larger datasets, and **we have reported experimental results on large OGB datasets with 40K+ samples (see Appendix E.2)** and our methods also outperform all baselines.
>
> 2) We present the runtime analysis of different methods in the public response. Please refer to our explanations in **Q1** of the **public response** for details.
>
> [a] Xu Hongteng, et al. Representing graphs via Gromov-Wasserstein factorization[J]. IEEE Transactions on Pattern Analysis and Machine Intelligence, 2022, 45(1): 999-1016.

---

> ### Comment · Area_Chair_vRuB · 2023-08-19
>
> Dear Reviewer,
>
> The authors have provided a comprehensive response to your review. Could you please confirm if it addresses your concerns? With the rebuttal deadline fast approaching, we would greatly appreciate your feedback before then.
>
> Thank you for your time and consideration.

---

> ### Comment · Reviewer_Cjiv · 2023-08-19
>
> I appreciate the response from the authors. After reading it and the other reviews, I plan to raise my score from 5 to 6.
>
> However, I would like to encourage the authors to clearly indicate the related appendix contents in the main text; I believe some readers would be similarly confused by the current form, like me. Furthermore, I would urge the authors to indicate the limitations in the next revision: 1. the current slow implementation, and 2. we cannot "select a step size as small as possible" in practice due to the time constraint.

---

> > ### Author Response · Authors · 2023-08-19
> > **Reply to Reviewer Cjiv**
> >
> > We would like to express our genuine gratitude for your thoughtful and detailed reviews and for considering our rebuttal. Sorry again for our missing indications of the related Appendix contents in our main texts. We will make these remedies and supplement the limitations that you have mentioned in our final version.
> >
> > Thank you once again for your time, efforts, and positive assessments. Your comments have been constructive and helpful in refining our research.

---

### Author Rebuttal · Authors · 2023-08-08

Thanks for all the constructive suggestions and comments concerning our paper from five nice reviewers. Here we provide our responses to some questions that are commonly asked by the reviewers.

> **Q1. Can you present the runtime comparison between your methods and compared baselines? Can you further analyze the time complexity of FGWMixup?**

We present the averaged efficiencies of different mixup methods and time spent on training vanilla backbones of each fold on PROTEINS and NCI1 datasets as follows. As Reviewer WT1n requires, we also include G-Mixup with GW graphon estimators (denoted as G-Mixup+GW) in the table.

|          | DropEdge | DropNode | G-Mixup | ifMixup | G-Mixup+GW | FGWMixup | FGWMixup*  |
|:--------:|:--------:|:--------:|:-------:|:-------:|:----------:|:--------:|:----------:|
| PROTEINS | 0.192    | 0.229    | 6.34    | 2.08    | 2523.78    | 802.24   | 394.57     |
| NCI1     | 0.736    | 0.810    | 10.31   | 5.67    | 9657.48    | 1711.45  | 637.41     |

|          | GCN    | GIN    | Graphormer | GraphormerGD  |
|----------|--------|--------|------------|---------------|
| PROTEINS | 47.13  | 52.68  | 2636.98    | 2371.31       |
| NCI1     | 209.28 | 338.72 | 2701.04    | 6175.99       |

We can observe that FGWMixup(\*) and G-Mixup+GW are slower than the other data augmentation methods. The main reason is that the complexity of calculating (F)GW distances between two graphs is cubic ($O(mn^2+nm^2)$) [a], where $m, n$ are the sizes of two graphs. Moreover, when calculating barycenters, we need an outer loop with T iterations and M graphs. In total, the time complexity of mixing up two graphs of size $n$ is $O(MTn^3)$. FGWMixup\* boosts the efficiency by enhancing the convergence rate and reducing the required iterations T (see Table 5 in Appendix E.1 for more details), whereas G-Mixup+GW will have to go over the whole dataset to calculate graphons, which is much more time-consuming than FGWMixup.

However, sacrificing complexity to pursue higher performance has been the recent trend of technical development, e.g. GPT-4. Moreover, **we believe that the current time complexity of FGWMixup\* is still acceptable compared with the time cost of model training (especially Graphormers) and our performance improvements**, as most compared graph augmentation methods cannot effectively enhance the model performance as shown in Table 1.

More importantly, in practice, the main computational bottleneck of (F)GW-based method is the OT network flow CPU solver in the current implementation based on the most widely used POT lib. In other words, GPU-based network flow algorithms have not been applied in current computation frameworks. Moreover, mini-batch parallelization is not yet deployed in POT. However, recent works [b, c] from NVIDIA have focused on accelerating network flow algorithms on GPU, which may probably be equipped on CUDA and allow a huge acceleration for GW solvers in the near future. **Hence, we firmly believe that the fact that our method is not yet optimized in parallel on GPUs is only a temporary problem.** Just as some other works (e.g., MLP, LSTM, etc.) that have brought enormous contributions in the past era, they are initially slow when proposed, but have become efficient with fast-following hardware supports.

> **Q2. Analysis of infeasibility of the single-loop FGW solver?**

First of all, we conduct an experiment to analyze the infeasibility of our single-loop FGW solver compared with the strict CG solver. We randomly select 1,000 pairs of graphs from PROTEINS dataset and apply the two solvers to calculate the FGW distance between each pair of graphs. The distances of the $i$-th pair of graphs calculated by the strict solver and the relaxed solver are denoted as $d_i$ and $d^{\*}_i$, respectively. We report the following metrics for comparison: i) MAE (mean absolute error of FGW distance, $\frac{1}{N}\sum |d_i - d^{\*}_i|$), ii) MAPE (mean absolute percentage error of FGW distance,  $\frac{1}{N}\sum \frac{|d_i - d^{\*}_i|}{d_i}$), iii) mean FGW distance given by the single-loop solver ($\frac{1}{N}\sum d^{\*}_i$), iv) mean FGW distance given by the strict CG solver ($\frac{1}{N}\sum d_i$), v) L2-norm of the difference between two transportation plan matrices (divided by the size of the matrix for normalization). The results are shown as follows:

| MAE            | MAPE           | mean_FGW | mean_FGW\* | T_diff          |
|----------------|----------------|----------|-----------|-----------------|
| 0.0126(0.0170) | 0.0748(0.1022) | 0.2198   | 0.2143    | 0.0006(0.0010)  |

We can observe that the MAPE is only 0.0748, which means the FGW distance estimated by the single-loop relaxed solver is only 7.48% different from the strict CG solver. Moreover, the absolute error is around 0.01, which is quite small compared with the absolute value of FGW distances (~0.21). We can also find that the L2-norm of the difference between two transportation plan matrices is only 0.0006, which means two solvers give quite similar transportation plans. All the results imply that **the single-loop solver will not produce huge infeasibility or make the estimation of FGW distance inaccurate**.

From another perspective, we do not think the infeasibility brings totally negative effects. As analyzed in our experiments (Table 1), FGWMixup* even sometimes performs better than FGWMixup. We explain this with the subtle infeasibilities introduced by the single-loop solver. By incorporating those infeasibilities, FGWMixup* may be able to **occasionally generate more diverse examples, which potentially enlarge the input space and bring opportunities to improve the generalizability of GNN models.**

[a] Gabriel Peyre et al., Gromov-Wasserstein Averaging of Kernel and Distance Matrices, ICML 2016

[b] https://on-demand.gputechconf.com/gtc/2017/presentation/S7370-hugo-braun-efficient-maximum-flow_algorithm.pdf

[c] https://mate.unipv.it/gualandi/talks/Gualandi_Aussois2020.pdf

---

### Comment · Area_Chair_vRuB · 2023-08-19

I would like to thank the authors for providing detailed responses to all referee reports, and I apologize that some of the referees have not responded to the rebuttals. I will account for that in my final recommendation.

---

> ### Author Response · Authors · 2023-08-19
> **Reply to Area Chair vRuB**
>
> Thank you so much for assisting us to ensure our rebuttals have been comprehensively considered during the discussion phase. Your support was a great help and is much appreciated.

---

### Decision · Program_Chairs · 2023-09-21

**Decision:**

Accept (poster)

**Comment:**

This paper proposes a novel graph mixup method called FGWMixup that considers the joint interaction between graph structures and signals. FGWMixup seeks a "midpoint" of source graphs in the Fused Gromov-Wasserstein metric space to generate augmented graph samples. The authors also introduce a relaxed FGW solver to improve efficiency.

The reviewers raised several valuable concerns, including the computational complexity (WT1n), the quality of the synthetic graphs (Ake3), comparisons to related work like G-Mixup (WT1n), and theoretical justifications (Cjiv). The authors provided comprehensive rebuttals addressing each point. They analyzed the runtime and complexity compared to baselines, shared qualitative examples demonstrating meaningful graph structures and signals, clarified differences from G-Mixup, and explained the theoretical propositions in more detail. The reviewers acknowledged the satisfactory responses in their follow-up comments.

After considering the paper, reviews, rebuttal, and discussion, we believe this work makes a valuable contribution as an innovative graph mixup approach. The experiments demonstrate effectiveness on multiple datasets and GNN architectures. The theoretical analysis also strengthens the methodology. While more work can be done to optimize efficiency and refine the graph generation process, the core ideas warrant acceptance.

In conclusion, acceptance is recommended for this paper as a poster presentation.